# An Open-Source Automatic Survey of Green Roofs in London using Segmentation of Aerial Imagery

Charles H. Simpson, Oscar Brousse, Nahid Mohajeri, Michael Davies, and Clare Heaviside

UCL Institute for Environmental Design and Engineering, The Bartlett School of Environment Energy and Resources, University College London, London, United Kingdom

**Correspondence:** Charles Simpson (charles.simpson@ucl.ac.uk)

**Abstract.**

Green roofs can mitigate heat, increase biodiversity, and attenuate storm water, giving some of the benefits of natural vegetation in an urban context where ground space is scarce. To guide the design of more sustainable and climate resilient buildings and neighbourhoods, there is a need to assess the existing status of green roof coverage and explore the potential for future implementation. Therefore, accurate information on the prevalence and characteristics of existing green roofs is needed, but this information is currently lacking. Segmentation algorithms have been used widely to identify buildings and land cover in aerial imagery. Using a machine-learning algorithm based on U-Net to segment aerial imagery, we surveyed the area and coverage of green roofs in London, producing a geospatial dataset (Simpson et al., 2023). We estimate that there was $0.23 \ km^2$ of green roof in the Central Activities Zone (CAZ) of London, $(1.07 \ km^2)$ in Inner London, and $(1.89 \ km^2)$ in Greater London in the year 2021. This corresponds to 2.0% of the total building footprint area in the CAZ, and 1.3% in Inner London. There is a relatively higher concentration of green roofs in the City of London, covering 3.9% of the total building footprint area. Test set accuracy was 0.99, with an f-score of 0.58. When tested against imagery and labels from a different year (2019), the model performed just as well as a model trained on the imagery and labels from that year, showing that the model generalised well between different imagery. We improve on previous studies by including more negative examples in the training data, and by requiring coincidence between vector building footprints and green roof patches. We experimented with different data augmentation methods, and found a small improvement in performance when applying random elastic deformations, colour shifts, gamma adjustments, and rotations to the imagery. The survey covers $1558 \ km^2$ of Greater London, making this the largest open automatic survey of green roofs in any city. The geospatial dataset is at the single-building level, providing a higher level of detail over the larger area compared to what was already available. This dataset will enable future work exploring the potential of green roofs in London and on urban climate modelling.

# 1   Introduction

In urban areas, green roofs (i.e. roofs deliberately covered in a growing substrate and living vegetation) can provide some of the benefits of ground-level green-space to health, well-being and the environment. Studies have examined the extent to which green roofs can directly reduce cooling energy demand and the risk of overheating in buildings (e.g. Castleton et al. (2010); Sailor et al. (2012); Sproul et al. (2014); Virk et al. (2015)), or can provide indirect benefits by decreasing the outdoor air temperature in hot weather with mixed results (e.g. Peng and Jim (2013); Virk et al. (2015); Cuthbert et al. (2022)). Green roofs have the potential to provide a range of benefits to humans, and to the wider ecological system in cities by providing habitats for wildlife (Filazzola et al., 2019; Hoeben and Posch, 2021), and can act as a carbon sink (Getter et al., 2009). Furthermore, green roofs may be able to contribute to the removal of air pollutants (Baik et al., 2012), and storm water retention (Mentens et al., 2006). Thus, green roofs are increasingly seen as an opportunity to improve health and well-being in urban environments, and as a part of climate mitigation and adaptation strategy. On the other hand, green roofs impose additional structural loads and costs, so are not always appropriate (Losken et al., 2018); in other cases solar panels or high-albedo roofs may be more appropriate. Cities worldwide have policies that encourage the use of green roofs through quantitative planning tools (The Ecology Consultancy, 2017).

Previous technical reports commissioned by the Greater London Authority (GLA) have investigated the area of green roofs in London (Table 1). The 2019 Living Roofs and Walls report (hereafter LRW2019), surveyed existing green roofs for the years 2016 and 2017 ( (European Federation of Green Roof and Green Wall Associations (EFB) and Livingroofs.org on behalf of the Greater London Authority, 2019; Livingroofs Enterprises Ltd, 2019)), although the methods are not publicly documented. The survey reports estimates for Greater London by local authority district (LAD) and for the Central Activities Zone (CAZ: a central area in London defined for planning purposes, see subsection 2.1 and Figure 1). In the London Plan Annual Monitoring Report (AMR), another report for the GLA, green roof areas in the CAZ were estimated based on aerial imagery for years 2013, 2015, and 2017 (Greater London Authority, 2021a, Key Performance Indicator 22, page 70) ranging from $1.75 \times 10^5 \ m^2$ in 2013 to over $2.9 \times 10^5 \ m^2$ in 2015. Lastly, an interactive map of green roofs in the CAZ is publicly available on the GLA website (Greater London Authority, 2014); it was produced in 2013/14 and is consistent with a green roof area in the CAZ of $1.75 \times 10^5 \ m^2$. Although these different estimates (Table 1) offer valuable information on recent green roof coverage in London they lack transparency about the methods used, there is a wide disagreement about the area of green roofs in the CAZ, and the full data are not publicly available for analysis.

Accurate, comprehensive, and open data documenting the location and area of green roofs can directly inform research into city-scale heat mitigation strategy and is useful for stakeholders such as urban planners, policy makers and research communities looking at urban heat mitigation and the added value of green spaces. However, there is a general lack of open data documenting the area and coverage of green roofs. In order to address this Wu and Biljecki (2021) applied a machine-learning algorithm to high-resolution satellite-imagery to identify green roofs and solar panels in a number of cities around the world, producing a ranking for which of the surveyed cities have the greatest coverage with green roofs and solar panels. London was not included in their survey.

**Table 1.** Previous estimates of green roof area in London. CAZ refers to a central area of London, see Subsection 2.1. Data from (European Federation of Green Roof and Green Wall Associations (EFB) and Livingroofs.org on behalf of the Greater London Authority, 2019) and (Greater London Authority, 2021a).

| Survey area | Source | Survey year | Green roof area ($10^5\ m^2$) | % of built area |
|---|---|---|---|---|
| CAZ | London Plan AMR 16 | 2013 | 1.75 | 1.5 |
| CAZ | London Plan AMR 16 | 2015 | 2.2 | 1.9 |
| CAZ | London Plan AMR 16 | 2017 | over 2.9 | 2.5 |
| CAZ | LRW2019 | 2016 | 1.5 | 1.3 |
| CAZ | LRW2019 | 2017 | 2.1 | 1.8 |
| Greater London | LRW2019 | 2016 | 11.0 | 0.43 |
| Greater London | LRW2019 | 2017 | 15.0 | 0.59 |

In this study, we identify green roofs from aerial imagery: this is a binary segmentation problem, as a single class needs to be identified from a background. Such algorithms process an image to output a binary mask identifying areas belonging to the target class. We used a fully convolutional neural network known as U-Net to segment the imagery: this type of neural network was originally designed for biomedical image segmentation (Ronneberger et al., 2015), but have since been applied in other research fields including remote sensing: for example to map roads (Ozturk et al., 2020), parking lots (Ng and Hofmann, 2018),

and green roofs (Wu and Biljecki, 2021) from imagery.

     Green roofs cover only a small proportion of the planar area of London, so in aerial imagery most pixels are not part of a green roof. This means that the classification problem is imbalanced, with the negative class being many times more numerous that the positive class. This can create problems with model training if gradient descent batches often do not contain any positive examples. In Wu and Biljecki (2021), the training polygons were restricted to areas with relatively higher concentrations of

green roofs and image tiles with no green roof were excluded; 1-5 $km^2$ of each of the 17 cities covered. Furthermore, the total number of examples for training is relatively low compared to many computer vision tasks, meaning that a computer vision model may be unable to generalise the appearance of green roofs; as such, data augmentation is thought to be key for achieving good segmentation performance. In the original U-Net paper, elastic deformations are applied to the training images, which makes the network learn to be invariant to these deformations without the need for all possible deformations to be present in

the data (Ronneberger et al., 2015), this is justified as soft tissues in medical images are often deformed in this way. In Ng and Hofmann (2018) (on which Wu and Biljecki (2021) is based), random rotations in units of $90^o$ and horizontal flips were applied to the images, in order enforce rotational independence to the classifier and reduce overtraining.

     In this study, we build on the machine-learning based method used by Wu and Biljecki (2021) for the segmentation of green roofs from remote-sensed imagery, improve the segmentation performance by including more negative examples, and

experiment with data augmentation methods. We thus provide a robust, open and documented dataset of the location and area of green roofs in London at the level of individual buildings (Simpson et al., 2023), filling a gap in publicly available data. This dataset has the greatest extent of its kind for any single city.

## 2  Data and Methods

### 2.1  Geographic Context and Data

Greater London is a region of England with an area of $1,570\ km^2$, which is divided into local authority districts (LADs), which are the 32 boroughs and the City of London. Inner London, with an area of $319\ km^2$ is defined by the Office for National Statistics; it comprises 14 LADs in the centre of London, roughly corresponding to the historic county of London (Office for National Statistics, b). The Central Activities Zone (CAZ) is the historic, governmental, and business centre of London defined by the GLA for planning purposes (Greater London Authority, 2021b, Policy SD4). The CAZ is contained

within Inner London but does not align with the LAD boundaries; it intersects with 10 LADs and has an area of $33.5\ km^2$. Lower super output areas (LSOAs) are areas with 1000-3000 residents defined for the purpose of census statistics: each LSOA is within exactly one LAD, and each LAD contains multiple LSOAs. In this article we use the LSOAs defined for the 2011 census (Office for National Statistics, a).

Using the Local Climate Zone typology (Stewart and Oke, 2012; Demuzere et al., 2019) as a reference, the built form of

Greater London is mostly classified as open lowrise. Inner London covers most of the area classified as open midrise and compact midrise, but also contains a large amount of open lowrise. The CAZ mainly covers the area of compact midrise in the centre and is therefore the most densely built part of London. Buildings in the CAZ, and especially the City of London, are more likely to be non-residential buildings. Figure 1 show the outlines of the LADs in Greater London and Inner London, and the outline of the CAZ.

Datasets described in this section are summarised in Table 2.

| Dataset | Type | Source | Explanation |
|---|---|---|---|
| Imagery 2019 and 2021 | $25\ cm$ RGB raster | Getmapping Inc. | Cloud free vertical aerial imagery mosaic. |
| OS VML | Vector | Ordance Survey | Large scale building outlines. Dated July 2021. |
| UKBuildings | Vector | Verisk Ltd. | Building footprints. Version 12, dated March 2022. |
| LSOA outlines | Vector | Office of National Statistics | Lower-super output areas from 2011 Census. |
| LAD outlines | Vector | Office of National Statistics | Local authority districts boundaries. |

**Table 2.** Input geospatial dataset summary.

The imagery used for segmentation comprised of raster images with red, green and blue bands from cloud-free mosaics of aerial imagery at $25\ cm$ horizontal resolution (from (Getmapping Plc., 2020) accessed under an academic license). Two sets of imagery were used, from 2019 and 2021. The imagery from 2021 was used as the primary dataset, with the imagery from 2019 providing an alternative dataset to test generalization. The collection dates for the imagery mosaic covering Greater London are

shown in Figure 2. The 2021 imagery covers $1706\ km^2$ of which $1558\ km^2$ was inside the Greater London boundary, while the 2019 imagery covers $1527\ km^2$ of which $1422\ km^2$ was inside the Greater London boundary.

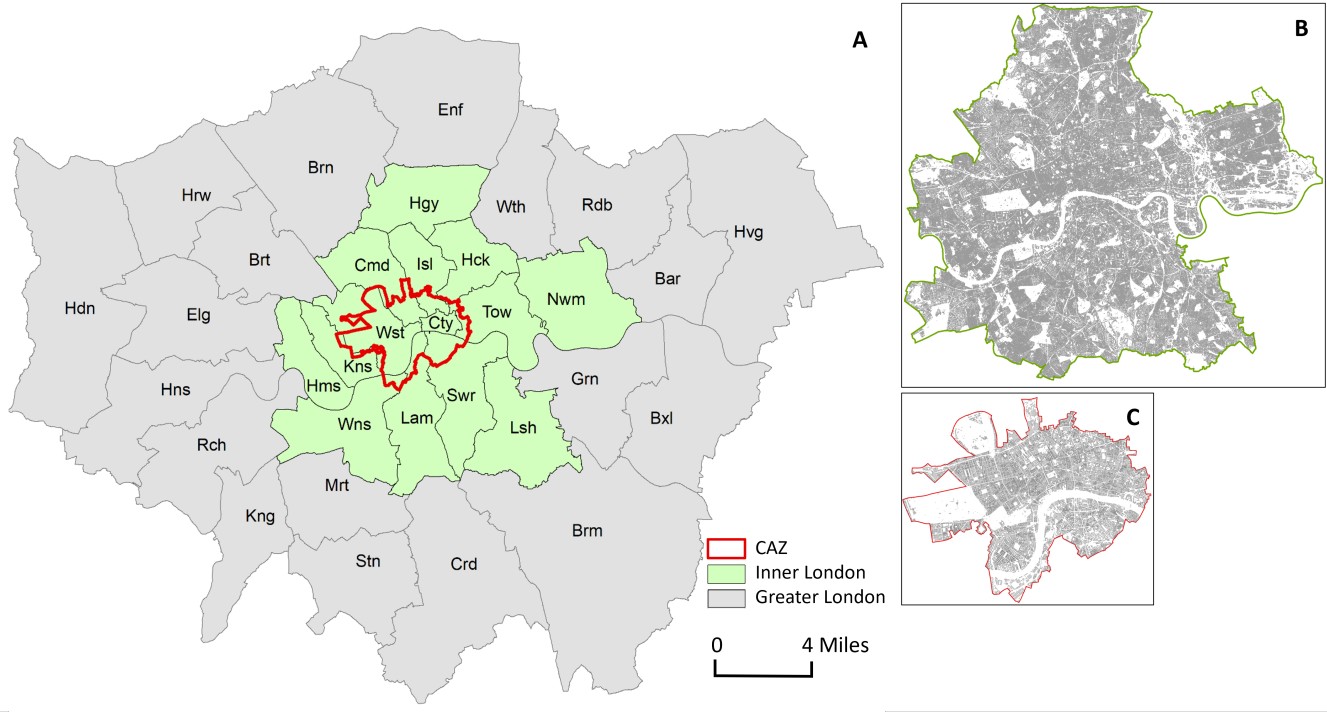

**Figure 1.** (A) Local authority districts in Greater London, identified by first three consonants of their names. Inner London is highlighted in green and the outline of the CAZ is shown in red. Zoomed in maps of (B) Inner London, and (C) the CAZ.

Two GIS datasets were used for building footprints. Ordnance Survey (OS) VectorMap Local (VML) (Ordnance Survey (GB), 2021) building footprints dated April 2019 were used in post-processing the segmentation, as inspection showed that outlines were more consistent with the aerial imagery, especially in cases of buildings with internal courtyards. UKBuild-ings (Verisk Analytics, Inc., 2022) building footprints were used for building counts, as it divides buildings into individual properties.

## 2.2 Segmentation pipeline

Our segmentation pipeline was based on that of Wu and Biljecki (2021), which is in turn based on Ng and Hofmann (2018). The key differences are as follows:

1. we used aerial imagery rather than satellite imagery,

2. our hand-labelled areas are distributed around the city rather than concentrated in a central area,

3. we focussed on fully surveying a single city rather than trying to cover many,

4. we experimented with additional data augmentation methods,

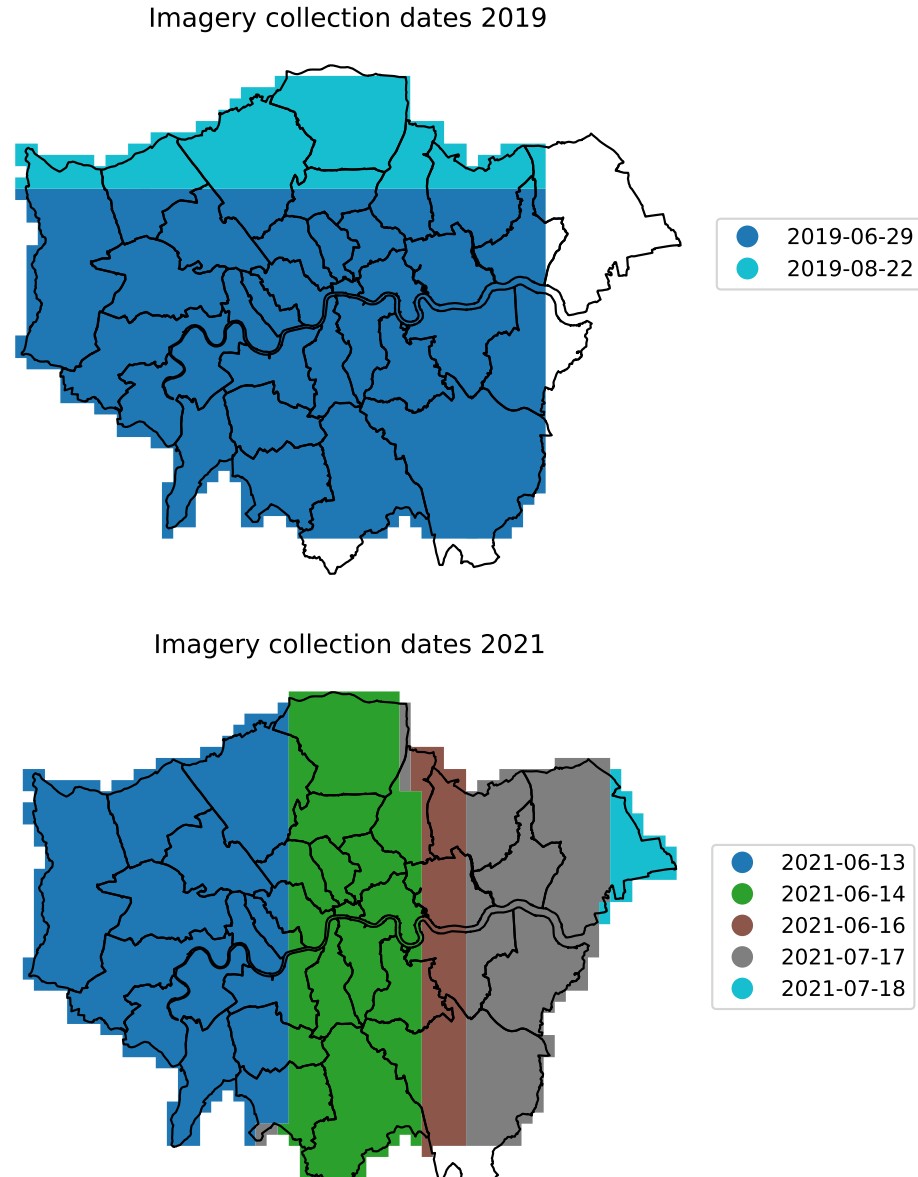

**Figure 2.** Collection dates for the aerial imagery mosaic covering Greater London. The primary imagery dataset used was that from 2021, while the 2019 data were used for comparison.

5. we implemented early stopping rather than training for a fixed number of epochs,

6. we tested different loss functions to handle the imbalanced nature of the problem,

7. we experimented with freezing the pretrained layers of the model,

8. we did not use morphological opening or closing to adjust the prediction raster,

9. we used building footprints provided by Ordnance Survey rather than OpenStreetMap for postprocessing,

10. we included tiles containing no positive examples in training.

All analysis and data management was performed using Python (Van Rossum and Drake, 2009). A general outline of the workflow is shown in Figure 3. The method is covered in more detail in the following subsections.

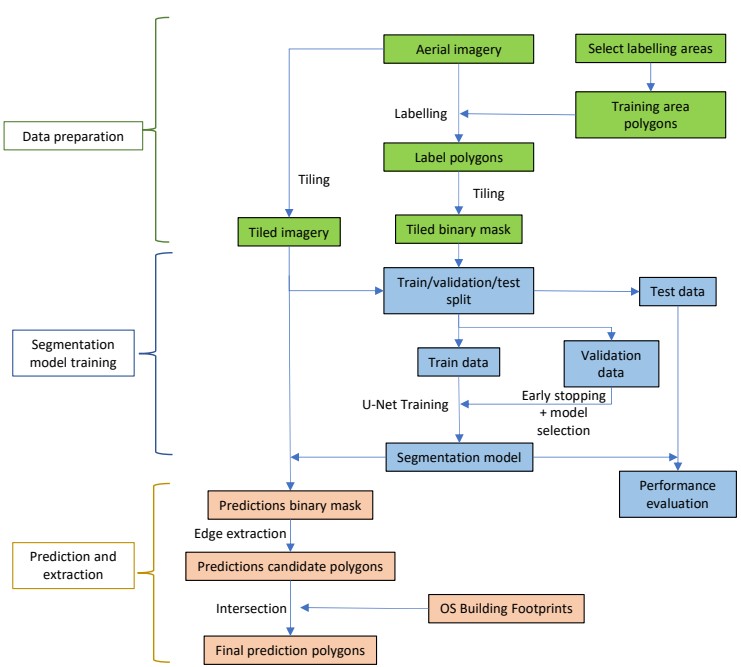

**Figure 3.** Workflow diagram of the overall segmentation pipeline.

## 2.3  Imagery and labelling

To identify the locations of green roofs and estimate their covered area, we trained our U-Net with training polygons from a sample area. The encoder layers of the U-Net produce compressed abstract representations of the image at different scales, by

repeatedly using convolution blocks followed by maxpool downsampling. The decoder layers apply upsampling and concatenation with convolution to produce a prediction with the same dimensions as the input image, combining information from the different scales provided by each encoder layer. The relationship between the image and the classification is learned from a set of labelled examples, hereafter referred to as training polygons. To produce training data, green roofs in the imagery were labelled by hand to provide input for model training. The training polygons and geospatial results are included in the data repository associated with this article to this article for reproducibility Simpson et al. (2023). We selected areas for labelling based on the OS 1 km grid reference system, so each grid square is $1 \ km^2$. Firstly, a $4 \ km^2$ area in the CAZ was selected, known to have a relatively higher concentration of green roofs: this was to ensure that there is sufficient representation of green roofs in the data. Secondly, to increase the diversity of the data, we selected a further $21 \ km^2$ distributed around Inner London without prior knowledge of the concentration of green roofs, aiming to represent each LAD and a variety of built forms (based on an LCZ map); these areas had much smaller amounts of green roofs. All grid references that were included are listed in Table A2 and mapped in Figure 4. Within the selected grid squares, every building in the imagery was inspected and green roofs were labelled by hand. Labelling was performed by drawing polygons using QGIS (QGIS Association, 2022); some examples of training polygons are shown in Figure 5. Labels were initially produced with reference to the 2019 imagery, and then were modified with reference to the 2021 imagery; the labels are different for the two datasets. In total, sample areas covered 7.8% of Inner London, resulting in $4.9 \times 10^4 \ m^2$ (in 2019) and $5.7 \times 10^4 \ m^2$ (in 2021) of green roofs labelled inside the CAZ, and $2.3 \times 10^4 \ m^2$ (in 2019) and $3.3 \times 10^4 \ m^2$ (in 2021) outside the CAZ.

Once trained we applied the U-Net to a larger area (the whole of Greater London) to map existing green roofs.

## 2.4 Performance metrics

Standard metrics were calculated to assess the validity of the segmentation model. Metrics were calculated from the final vector layers, after all processing steps. The metrics are listed in Table 3. Accuracy, intersection-over-union (IoU), precision, recall, and F-score all range from 0 to 1, where 1 represents an ideal classifier. F-score is a more appropriate measure of the overall validity of a model for imbalanced classification than accuracy. As well as calculating these metrics, we examined examples of poor segmentation performance to understand the failure modes of our segmentation method.

## 2.5 Segmentation algorithm

The imagery was broken into $256 \times 256$ pixel tiles at a scale of $0.25 \ m$ per pixel in the OSGB36 coordinate reference system. Pixel values were normalized to match ImageNet during training and prediction. We refer to areas labelled with no green roof as negative, and those labelled with any green roof as positive. All tiles within the hand-labelled areas were used. Negative pixels (i.e. those without green roof) were more numerous than positive pixels, and fully-negative tiles were more numerous than positive tiles. To include negative-only tiles (which are far more numerous) while ensuring that enough batches would contain positive examples, we experimented with two resampling methods during training: over-sampling positive tiles by repetition, or random sampling with replacement of the negative-only tiles.

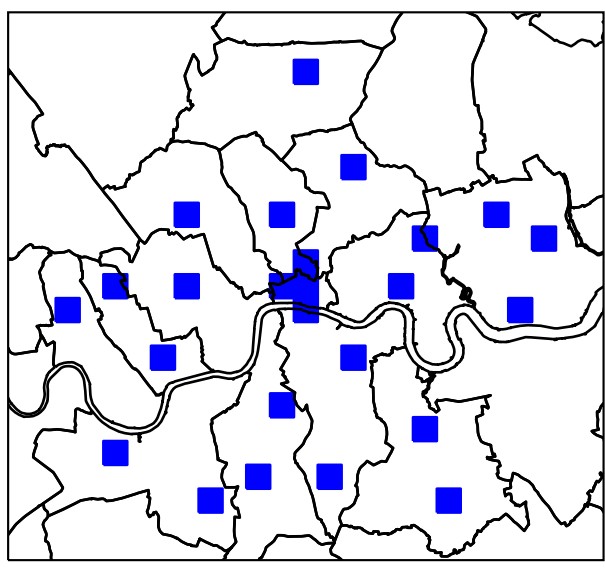

**Figure 4.** Map of hand labelled areas.

**Table 3.** Classification performance metrics calculated in this study.

| Metric | Definition |
| --- | --- |
| Accuracy | Proportion of area correctly classified |
| True positive (TP) | Area correctly classified as positive. |
| False positive (FP) | Area incorrectly classified as positive. |
| True negative (TN) | Area correctly classified as negative. |
| False negative (FN) | Area incorrectly classified as negative. |
| Intersection over union (IoU) | TP/(TP+FP+FN) |
| Precision | TP/(TP+FP) |
| Recall | TP/(TP+FN) |
| F-score | harmonic mean of precision and recall |

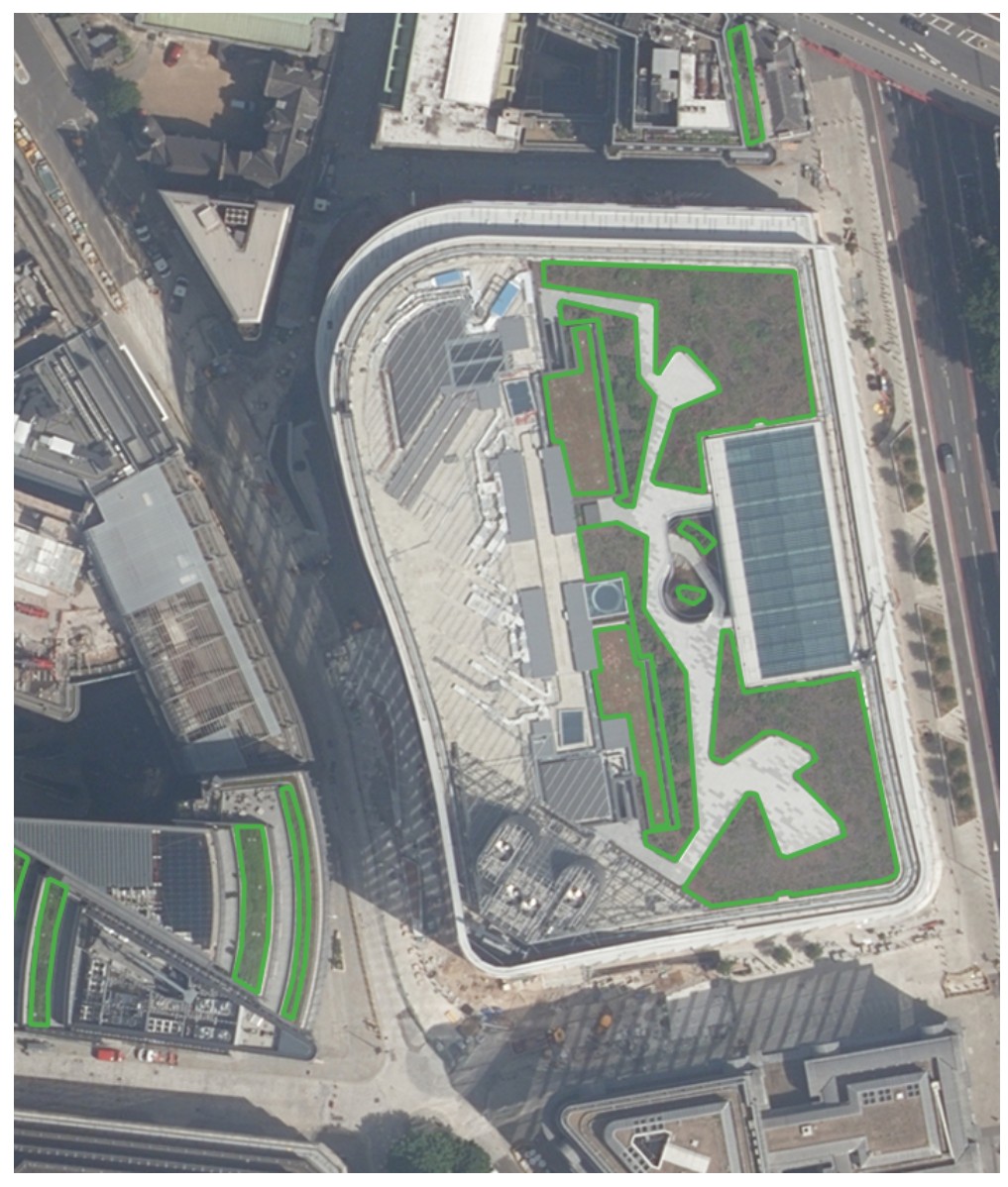

**Figure 5.** Example of training polygons. The area outlined in green was manually identified as being a green roof in this aerial imagery. Imagery ©GetMapping Plc. Image location is shown in Figure A3.

Transfer learning refers to the practice of transferring models or parts of models between different learning tasks - in this case from a well-known image classification task to our segmentation task. Ng and Hofmann (Ng and Hofmann, 2018) used transfer learning to mitigate the small number of training examples; the U-Net encoder is replaced with a ResNet50 trained on the ImageNet dataset (He et al., 2016; Deng et al., 2009), an example which we follow. Transfer learning can improve performance and reduce the required training resources as the model will have already learned to extract features from images that are generally informative. Freezing refers to the choice to not update the pre-trained part of the model during training, which can beneficial as it can massively reduce the compute time required for training: we experimented with freezing pre-trained layers.

The algorithm was implemented in PyTorch (Paszke et al., 2019). The model was trained using the Adam optimiser (Kingma and Ba, 2014), an optimiser that dynamically adjusts learning rates for each model parameter, making training less dependent on the global learning rate and therefore reducing required training resources.

Rather than training the model for a fixed number of epochs, we implemented early stopping. Early stopping refers to stopping training when validation performance ceases to improve. This reduces the required training resources and can effective at reducing overfitting. Training was stopped if the mean validation loss in the past five epochs was greater than that of the five epochs before.

It was not practical to test all combinations of hyperparameters, loss functions, augmentations etc. in a grid search. We therefore optimised each choice one-by-one: first loss function, then learning rate, augmentations, and freezing. Testing data were not used for training method tuning, and were only processed after the hyperparameters were finalised.

Four-fold cross-validation was performed; as required computational resources grows with the number of folds, we decided four was a good compromise between testing performance thoroughly and limiting resource usage. The hand-labelled tiles were split into five sets, of which one was reserved as the test dataset. The random split was performed separately for positive and fully negative tiles to ensure all splits contained both positive and negative examples. For each fold, training was performed with 3 of these sets, and validation with one set. This is to demonstrate that good performance is not unique to a particular random split of training and validation data, and therefore tests the ability of the model to generalise. To reduce resource requirements, optimisation of the training method was performed by maximising validation F-score using the first fold only, with only the final selected configuration being cross-validated. The test dataset remained unseen to all models during training, and was not used for choosing the optimal configuration, allowing for a good estimate of out-of-sample performance.

Cross-entropy loss, Lovasz loss, F-measure loss, and Focal loss functions were tested: Lovasz is intended as a surrogate for the intersection-over-union measure (Berman et al., 2018), whereas focal loss is intended to give greater weight to hard-to-classify examples during training (Lin et al., 2017). The cross-entropy and focal loss functions were weighted by the inverse frequency of the classes to account for class imbalance. In the final selected model configuration this resulted in the positive class having a weight of $10^2$ times that of and the negative class. Focal loss has a parameter gamma which adjusts the importance of different parts of the loss distribution, and different values were tested. The F-measure loss approximates the F1-score in a way that is differentiable, and leads to training that balances precision and recall without the need for weighting

(Pastor-Pellicer et al., 2013). Learning rate, loss function, and data augmentation methods were tested. The hyperparameters tuned, and hyperparameter values used for the final classification, are listed in Table A1.

A key part of the U-Net methodology is data augmentation Ronneberger et al. (2015) - a process wherein distortions or transformations are applied to the training data to increase robustness when training data is scarce. Augmentation can reduce overfitting, a process wherein a model memorizes certain features of the training dataset that do not generalise out-of-sample (Shorten and Khoshgoftaar, 2019). During training, augmentations were applied to the imagery tiles, and correspondingly to the label masks. Augmentation was applied randomly and independently each training epoch, with equal probability to positive and negative tiles. We experimented with flipping images in both planes, applying 90° rotation, applying random crops, randomly shifting the colours, randomly applying gamma adjustment, and randomly adjusting the sharpness of the imagery. We also experimented with applying elastic distortions. All augmentations used the Albumentations library (Buslaev et al., 2018).

## 2.6 Post-processing

Predicted segmentation masks were generated from the trained model using the same tiling method as used for training. The same prediction probability threshold of 0.5 was applied across the whole domain.

From the binary masks produced by the segmentation algorithm, we extracted green roof candidate polygons. Tiling resulted in zero-width gaps between polygons where as green roof straddled two tiles, this was closed by simply taking the union of these polygons. The intersection was then taken between the candidate polygons and the OS VML building footprints, to remove any candidate polygons that did not intersect with a building footprint. This process helped to reduce the false positive rate because the segmentation algorithm can incorrectly identify ground-level green cover as a green roof. We found it convenient to apply simplification of the prediction polygons using the iterative end-point fit algorithm and a threshold of $0.25\ m$, then to drop polygons smaller than $10\ m^2$, which had little impact on segmentation performance but reduced the file size (simplify routine implemented in Gillies et al. (2007–)). In previous work (Ng and Hofmann (2018); Wu and Biljecki (2021)), morphological opening and closing have been used on the raster classification masks as a post-processing step: these are filters that remove small isolated positive areas and fill in small negative areas respectively. But we found that these morphological operations decreased segmentation performance so we did not use them. The post-processed segmentation results were spatially joined with the UKBuildings layer in order to identify which individual buildings have green roofs, and so calculate the number of buildings covered.

## 2.7 Testing generalisation

As an additional test of out-of-sample performance, we included a second imagery dataset from a different year; the primary imagery was for the year 2021, the alternative dataset for the year 2019. First, training was performed using the 2021 imagery and labels, using k-fold cross validation to test the sensitivity of the performance to the train-test split. This model was tested against imagery and labels from the same year (2021) but also from an earlier year (2019). We compared the pixel-value distributions of the roof selected between these datasets. Further, we trained a single model using the 2019 imagery and labels, with exactly the same data split (i.e. the same geographic locations of tiles) as the first fold of the primary model; this model

was used to provide a benchmark for the performance of the primary model by testing against both 2021 and 2019 test data. Model design optimisation was performed only with the 2021 imagery and labels.

## 2.8 Area estimates

To estimate area of green roof in each geographic area, the polygons of green roof area identified by the segmentation are spatially intersected with the polygons of the geographic area. The same process is used with the building footprints to estimate building footprint area. All area calculations were applied in the OSGB36 / EPSG:27700 coordinate projection. Area projections are scaled up by the recall of the model, based on the assumption that a fixed proportion of each green roof is missed by the model. Not doing so would lead to an underestimation of the green roof area.

To estimate the change in green roof area between 2019 and 2021, we performed the geometric set difference between the 2021 and 2019 prediction polygons, with the 2019 prediction polygons buffered by 5 m to allow for errors in segmentation.

## 3 Results

### 3.1 Segmentation performance

The performance statistics averaged across folds for green roof identification are given in Table 4; performance statistics for all folds are given in Table A4, and the full confusion matrix in A3. Table 5 gives the same statistics calculated in terms of building counts rather than area; with performance statistics for all folds in Table A3 and the full confusion matrix in A5 Table 6 compares the performance of models trained on 2019 and 2021 imagery and labels, and tested against both 2019 and 2021 imagery and labels.

Results of the hyperparameter search are shown in Table A1. The best performance was found with the F-measure loss (F-score improvement 0.3), which may be because the class imbalance is large. We found that the augmentation which provided the greatest improvements in performance were the non-destructive transformations (flips and rotations) which provided an F-score improvement of 0.10 versus no augmentations, the effect of the other augmentations (elastic tranformation, colour shift, random gamma adjustment) were smaller, improving F-score by only a further 0.03. We found that over-sampling the positive tiles was more effective than not including any negative-only tiles, including all tiles without resampling, or under-sampling negative tiles. We experimented with the proportion of positive tiles to be achieved by resampling, and found the best results when 50% of tiles contained positive pixels. Training was roughly two times faster per epoch with the pre-trained part of the model frozen, so augmentation experiments were performed with it frozen, when the best combination was found training was repeated with the model un-frozen but this did not lead to an increase in F-score. We found that the building-intersection step increased testing precision by 0.05 on average across the folds for the 2021 testing dataset and 0.11 for the 2019 testing dataset with no effect on recall, showing that across the building-intersection step plays an important role in suppressing false positives. Figure 6 shows the distribution of colours in the predictions for the two imagery sets: generally true positives, false positives, and false negatives have strongly overlapping colour distributions which are similar between the two imagery sets.

**Table 4.** Performance metrics for the green roof identification method, calculated based on area. For the full set of statistics for all folds see Supplementary Table A4

|  | Accuracy | IoU | Precision | Recall | F-score |
|---|---|---|---|---|---|
| Dataset |  |  |  |  |  |
| Testing 2021 | 0.9925 | 0.3970 | 0.6505 | 0.5046 | 0.5683 |
| Testing 2019 | 0.9921 | 0.3707 | 0.5072 | 0.5793 | 0.5408 |
| Validation | 0.9923 | 0.4666 | 0.7148 | 0.5733 | 0.6363 |
| Training | 0.9941 | 0.5773 | 0.7374 | 0.7266 | 0.7320 |

**Table 5.** Performance metrics for the green roof identification method, calculated based on building counts. For the full set of statistics for all folds see Supplementary Table A6

|  | Accuracy | IoU | Precision | Recall | F-score |
|---|---|---|---|---|---|
| Dataset |  |  |  |  |  |
| Testing 2021 | 0.9924 | 0.4128 | 0.5721 | 0.5972 | 0.5844 |
| Testing 2019 | 0.9892 | 0.3256 | 0.3934 | 0.6536 | 0.4912 |
| Validation | 0.9920 | 0.4516 | 0.6024 | 0.6433 | 0.6222 |
| Training | 0.9925 | 0.4444 | 0.5404 | 0.7146 | 0.6154 |

**Table 6.** Comparison of test dataset performance model trained on 2019 imagery and labels with the model trained on 2021 imagery and labels.

|  | Accuracy | IoU | Precision | Recall | F-score |  |
|---|---|---|---|---|---|---|
| Dataset |  |  |  |  |  |  |
| Trained on 2019 |  |  |  |  |  |  |
| Testing 2021 | 1 | 0.9921 | 0.3682 | 0.6379 | 0.4655 | 0.5382 |
| Testing 2019 | 1 | 0.9909 | 0.3580 | 0.4580 | 0.6212 | 0.5273 |
| Trained on 2021 |  |  |  |  |  |  |
| Testing 2021 | 1 | 0.9929 | 0.4134 | 0.6801 | 0.5131 | 0.5849 |
| Testing 2019 | 1 | 0.9923 | 0.3782 | 0.5216 | 0.5790 | 0.5488 |

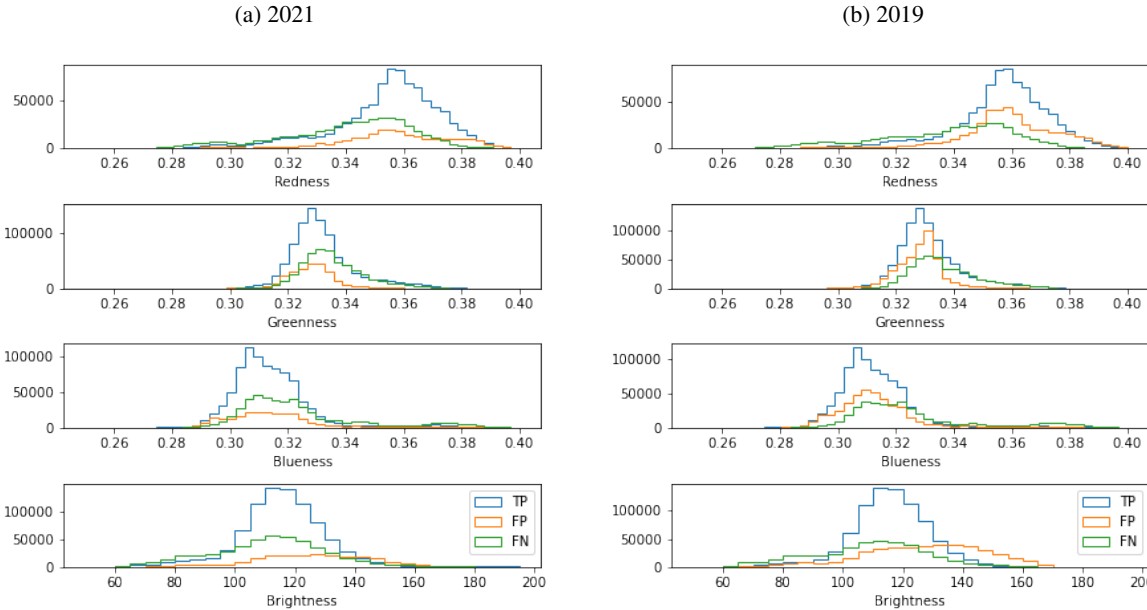

(a) 2021          (b) 2019

**Figure 6.** Colour and brightness of pixels in false positive (FP), true positive (TP), and false negative (FN) groups for (A) 2021 imagery and labels, (B) 2019 imagery and labels. The model was trained on 2021 imagery and labels.

## 3.2 Distribution of green roofs

Table 7 gives estimates for LADs in Inner London, Table 8 for Outer London, and Table 9 for aggregated areas. *Proportion of total building footprint area* means the total green roof area divided by the total building footprint area including all buildings not only those with green roofs. *Proportion of area* means the total green roof area divided by the area of the geography (LAD, CAZ, or Inner London). *Proportion of buildings* means the count of buildings with any green roof divided by the count of all buildings. *Proportion of buildings by area* means the building footprint area of buildings that have any green roof divided by the total building footprint area. *Mean coverage* means the total area of green roof divided by the total footprint area of buildings that have any green roof.

We estimate that the CAZ contained $2.27 \times 10^5 \ m^2$ of green roof on the dates of imagery collection (Summer 2021). Green roof area estimates for each LAD in Greater London, and LSOA in Inner London, are mapped in Figure 7. Most (58%) LSOAs contain no green roofs, and the maximum proportion of building footprint area covered by green roofs in any LSOA is 38%. We estimate that between 2019 and 2021, green roof area increased by $1.6 \times 10^4 \ m^2$ in the CAZ, $6.7 \times 10^4 \ m^2$ in Inner London, and $1.5 \times 10^5 \ m^2$ in Greater London.

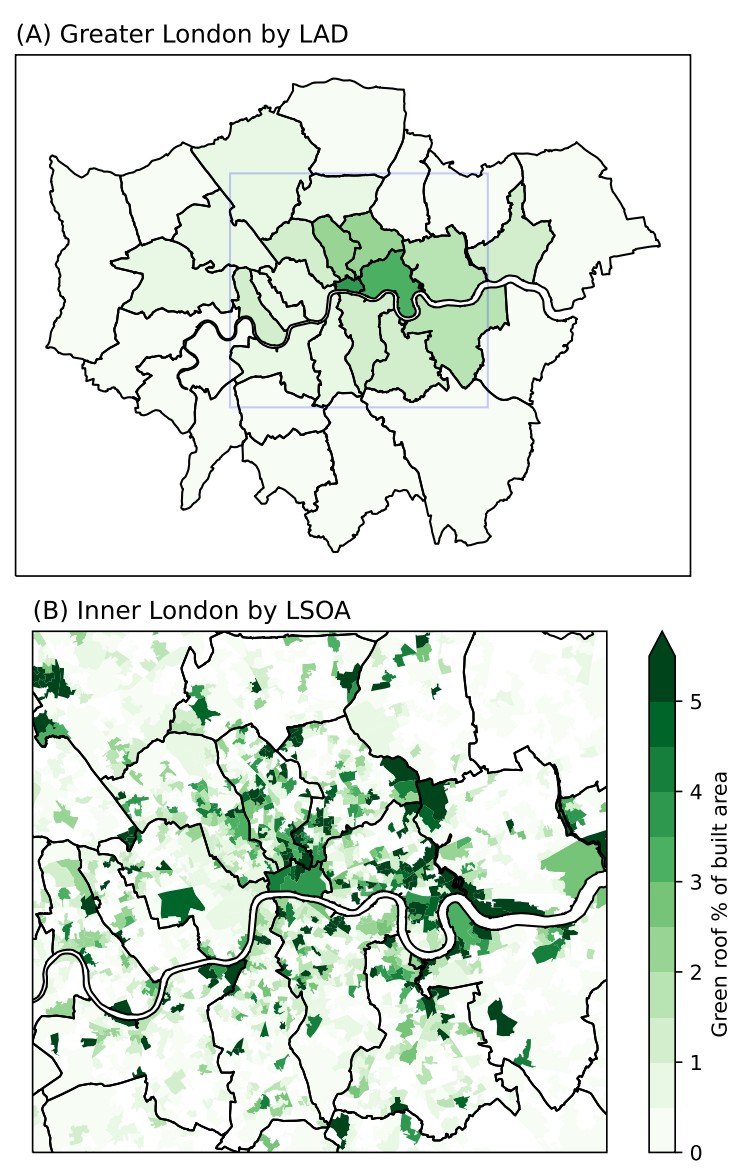

**Figure 7.** Area of green roof identified in (A) LADs and (B) LSOAs as a fraction of total building footprint area.

**Table 7.** Table of estimated green roof area for each LAD in Inner London in 2021.

| | Green roof area ($10^4\ m^2$) | Proportion of total building footprint area (%) | Proportion of geographic area (%) | Proportion of buildings (%) | Proportion of buildings by area (%) | Mean coverage (%) |
|---|---|---|---|---|---|---|
| City of London | 6.1 | 3.9 | 2.1 | 7.8 | 25.6 | 15.3 |
| Tower Hamlets | 15.6 | 3.1 | 0.8 | 1.7 | 11.4 | 27.7 |
| Islington | 10.0 | 2.2 | 0.7 | 1.0 | 6.7 | 33.2 |
| Hackney | 9.8 | 2.0 | 0.5 | 1.0 | 7.3 | 27.9 |
| Newham | 12.4 | 1.6 | 0.3 | 0.3 | 6.2 | 25.8 |
| Camden | 7.0 | 1.3 | 0.3 | 1.1 | 7.9 | 16.0 |
| Southwark | 8.4 | 1.2 | 0.3 | 0.7 | 6.0 | 20.7 |
| Hammersmith and Fulham | 5.4 | 1.2 | 0.3 | 0.5 | 9.2 | 12.7 |
| Lewisham | 7.9 | 1.1 | 0.2 | 0.4 | 3.8 | 29.8 |
| Wandsworth | 6.9 | 0.9 | 0.2 | 0.3 | 4.9 | 18.2 |
| Lambeth | 6.1 | 0.9 | 0.2 | 0.4 | 3.8 | 22.9 |
| Westminster | 5.3 | 0.8 | 0.2 | 1.1 | 7.7 | 10.0 |
| Haringey | 4.2 | 0.6 | 0.1 | 0.2 | 1.9 | 34.1 |
| Kensington and Chelsea | 1.8 | 0.5 | 0.1 | 0.4 | 2.8 | 17.2 |

## 4 Discussion

### 4.1 Segmentation Performance

The segmentation model achieves a high level of accuracy (0.99). Precision and recall based on area for the 2021 testing dataset are 0.65 and 0.50 respectively, with an F-score of 0.57 (Table 4). Based on counts of buildings instead, precision is lower (0.57) and recall is higher (0.60). This indicates that the model is effective at identifying green roofs, and that many of the false positives are small areas on buildings with no green roof.

Given that the survey covers such a large and diverse area, and the green roof fraction is low in many areas, it is important

to consider the false positive rate. Tables A3 and A5 suggest that we expect 0.3% of the built area to be incorrectly identified as green roof, which is comparable to the green roof area in some districts that have very little green roof e.g. Waltham Forest, but small in areas with more green roof.

Inspection of false negatives in the results showed that many pixels classified as false positives and false negative are at the edges of green roofs. In the k=1 validation dataset 31% of the false positive area is within a $1\ m$ buffer of the hand-labelled

polygons, and 34% of false negative area is within a $1\ m$ buffer of the predictions. This indicates that the dataset is good at identifying green roofs, but that there are often inconsistencies at the edge of the green roof between the human labeller and

**Table 8.** Table of estimated green roof area for each LAD in Outer London in 2021.

| | Green roof area ($10^4\ m^2$) | Proportion of total building footprint area (%) | Proportion of geographic area (%) | Proportion of buildings (%) | Proportion of buildings by area (%) | Mean coverage (%) |
|---|---|---|---|---|---|---|
| Greenwich | 12.3 | 1.6 | 0.3 | 0.3 | 5.3 | 30.3 |
| Barking and Dagenham | 6.9 | 1.1 | 0.2 | 0.3 | 6.1 | 17.4 |
| Brent | 7.3 | 0.8 | 0.2 | 0.3 | 3.4 | 22.8 |
| Barnet | 7.9 | 0.7 | 0.1 | 0.2 | 3.0 | 21.9 |
| Ealing | 6.6 | 0.6 | 0.1 | 0.2 | 2.5 | 23.7 |
| Richmond upon Thames | 3.1 | 0.4 | 0.1 | 0.2 | 2.1 | 20.8 |
| Hounslow | 4.0 | 0.4 | 0.1 | 0.2 | 2.9 | 14.9 |
| Hillingdon | 5.4 | 0.4 | 0.0 | 0.2 | 4.5 | 9.5 |
| Waltham Forest | 2.8 | 0.4 | 0.1 | 0.1 | 2.5 | 16.1 |
| Kingston upon Thames | 2.1 | 0.4 | 0.1 | 0.2 | 2.9 | 12.6 |
| Harrow | 2.7 | 0.3 | 0.1 | 0.2 | 1.5 | 22.3 |
| Sutton | 2.4 | 0.3 | 0.1 | 0.1 | 1.9 | 17.9 |
| Enfield | 3.7 | 0.3 | 0.0 | 0.1 | 1.8 | 18.8 |
| Merton | 2.1 | 0.3 | 0.1 | 0.2 | 1.3 | 23.8 |
| Croydon | 3.8 | 0.3 | 0.0 | 0.1 | 1.9 | 16.4 |
| Havering | 3.0 | 0.3 | 0.0 | 0.1 | 2.2 | 13.2 |
| Bexley | 2.2 | 0.2 | 0.0 | 0.1 | 1.4 | 16.0 |
| Bromley | 2.9 | 0.2 | 0.0 | 0.1 | 1.7 | 13.3 |
| Redbridge | 0.8 | 0.1 | 0.0 | 0.1 | 1.2 | 7.8 |

**Table 9.** Table of estimated green roof area for the CAZ, Inner London, and Greater London in 2021.

| | Green roof area ($10^4\ m^2$) | Proportion of total building footprint area (%) | Proportion of geographic area (%) | Proportion of buildings (%) | Proportion of buildings by area (%) | Mean coverage (%) |
|---|---|---|---|---|---|---|
| CAZ | 22.7 | 2.0 | 0.7 | 2.5 | 14.7 | 13.8 |
| INNER | 106.8 | 1.3 | 0.3 | 0.6 | 6.3 | 21.2 |
| TOTAL | 188.8 | 0.7 | 0.1 | 0.3 | 3.8 | 19.7 |

the model. A similar observation was made by Wu and Biljecki (2021) when discussing the relative difficulty of segmenting green roofs compared to solar panels, which have more well-defined edges in imagery. It may be that performance is limited by the consistency of the human labelling rather than the efficacy of the machine learning algorithm.

Differences in precision and recall in the test datasets between folds are small (see Table A7), showing that the performance is not unique to a particular train-test split and demonstrating the appropriateness of the model. It is possible that performance might be improved generally by labelling more data thus increasing the size of the training dataset, or through another method of data augmentation that was not explored.

     Comparing the performance of the same model (trained on 2021 imagery and labels) for the two testing datasets (2021
versus 2019), precision was lower for the 2019 dataset (from 0.57 to 0.39) but recall was higher (from 0.59 to 0.65) (Table 4). This means that with the alternative imagery dataset the model tends to include more a higher proportion of spurious green roofs. The difference in precision is greater when calculated in terms of building counts rather than area (0.59 to 0.39) (Table 5), suggesting that the additional false positives take the form of small areas on buildings without real green roofs. Imagery in the alternative set was completely unseen during training and optimisation. However, as Table 6 shows, performance is just as
good or better than a model trained on the 2019 images and labels. This demonstrates model can generalise to unseen imagery, although with some loss of precision.

     The IoU score for the testing dataset averaged across folds (0.397, Table 5) of our segmentation model is similar than that reported in Wu and Biljecki (2021) (0.396, see their sec 3.2.2). However, Wu and Biljecki (2021) did not include fully negative tiles in their training or validation; excluding fully negative tiles from our validation would increase the IoU by reducing false
positives. Wu and Biljecki (2021) covered a total of 2217 $km^2$ across 12 cities, with the largest being 302 $km^2$ in Las Vegas, Nevada; our survey covered 1558 $km^2$, making ours the largest survey of green roofs in a single city.

     While performance was generally good as measured by the performance metrics, we collected some examples of poor classification performance: Figure 8 shows some examples of false positives and Figure 9 false negatives. Shadows in the aerial imagery was a cause of both false positives and false negatives, for example Figure 8a as well as Figures 9a and 9b. This
could be because the shapes and colours are simply less distinct in shadow, but there are also few examples of this to learn from in the training data. The visual texture of roofs may be a source of false positives, for example Figures 8b, 8c, and 8d have a similar rough texture to the green roof in Figure 9d. Sedum-based green roofs often have a red-brown hue, meaning that another source of false positives are roofs with a similar red-brown hue as seen in Figures 8e and 8f; Figure 6 shows that true positives and false positives have strongly overlapping colour distributions, so it would not be possible to improve performance simply
by selecting certain colours. It could be that relatively small variations in colour lead to the misclassification, but we found that augmentations in gamma and colour only slightly improved performance. Multi-spectral imagery could help deal with variations in vegetation colour. However, multi-spectral aerial imagery is collected more rarely and is less available; satellite multi-spectral imagery is available but resolution is poorer. Therefore, visible-spectrum aerial imagery has some practical advantages over multi-spectral imagery. Combining layers of multi-spectral imagery at lower resolution with aerial-imagery is
technically challenging, but could be effective for this task. Sometimes part of a green roof is correctly identified, but patches are missed, as in Figures 9c and 9d.

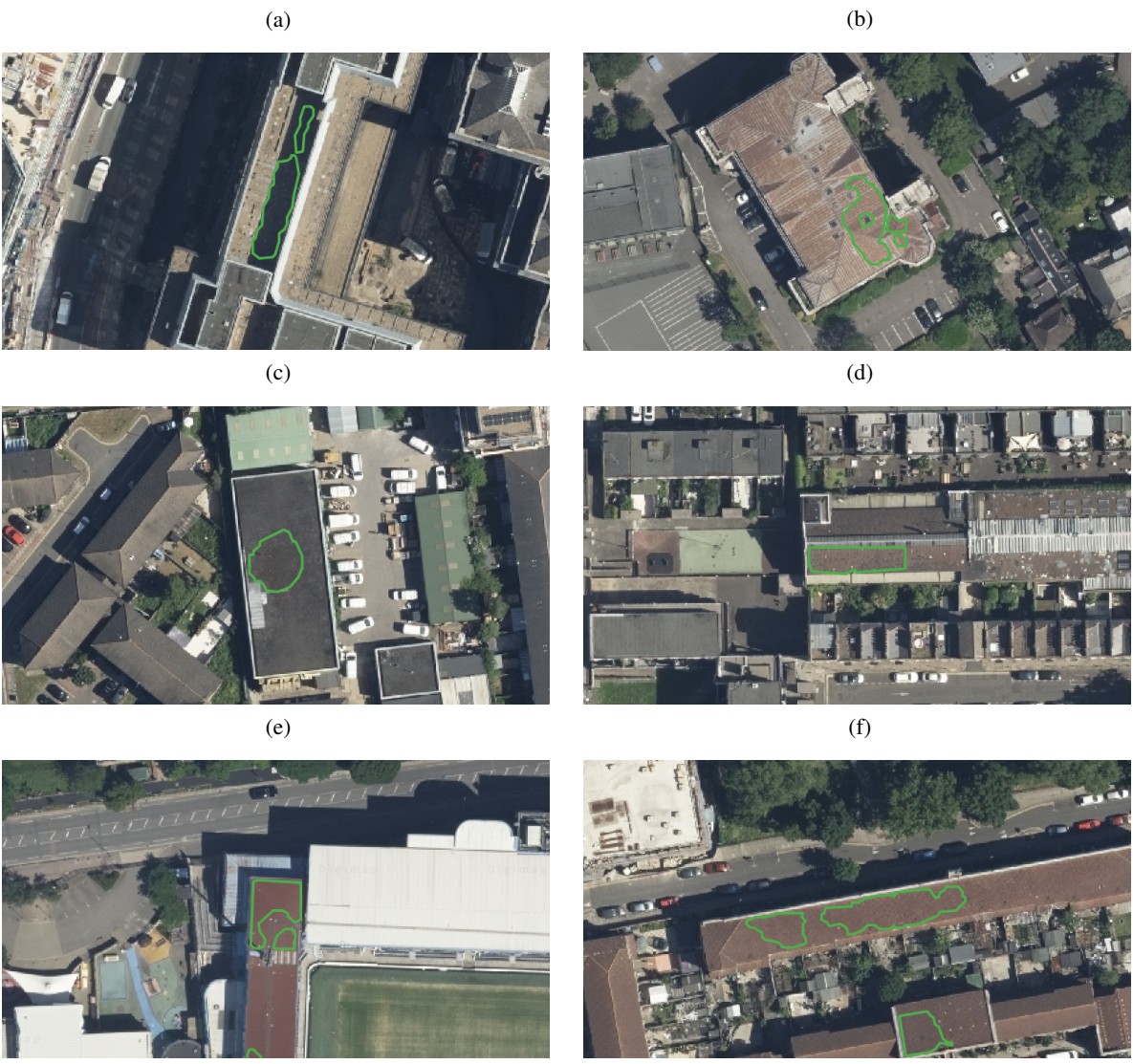

**Figure 8.** Examples of false positive classification in the 2021 imagery. The green outline shows the area identified as green roof by the model. Imagery ©GetMapping Plc. Image location is shown in Figure A3.

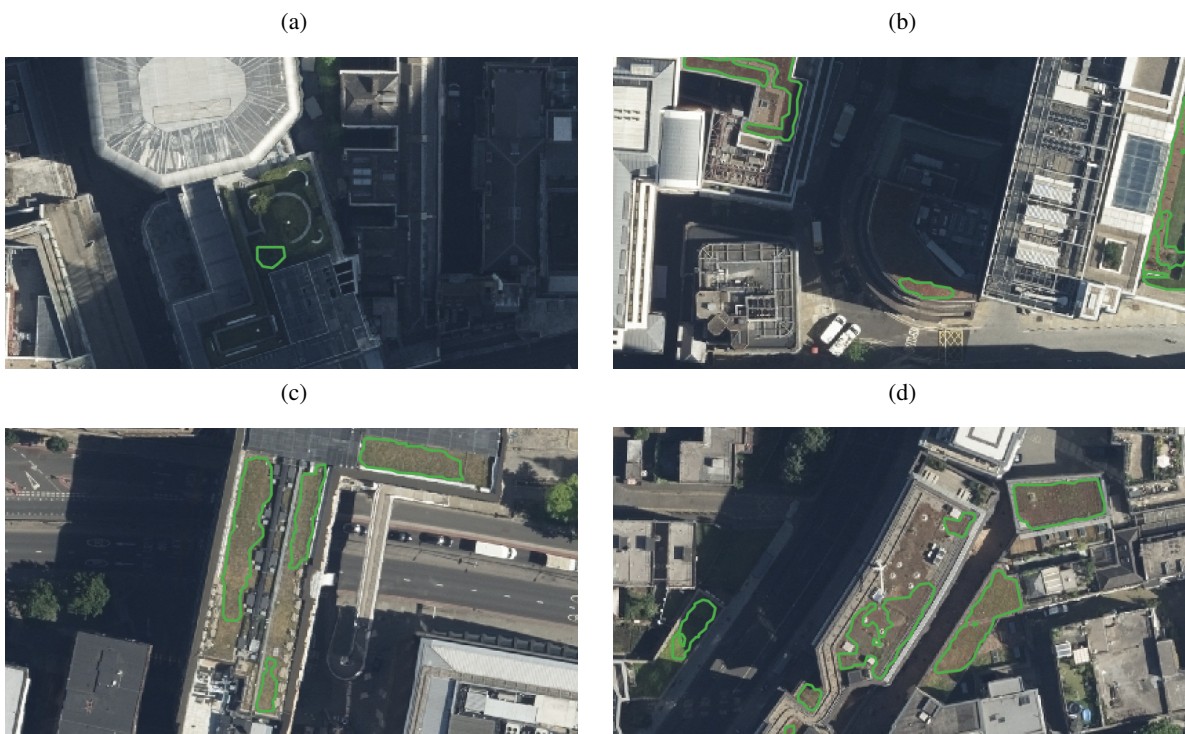

(a)          (b)

(c)          (d)

**Figure 9.** Examples of false negative classification in the 2021 imagery. The green outline shows the area identified as green roof by the model. Imagery ©GetMapping Plc. Image location is shown in Figure A3.

This method can be applied to other cities, and we have explored how the segmentation methods can be improved. While previous similar studies had included in training only tiles which contain positive examples, we found that including a large number of negative examples was very effective at suppressing the false positive rate in unseen areas despite increasing the class imbalance, and would recommend this approach in general. We observed improvement in segmentation performance by application of data augmentations, which we can recommend for future similar studies. The F-measure loss was also particularly effective for this problem as the segmentation classes are so imbalanced.

### 4.2 Limitations

As is clear from this study, automatic methods are scalable, allowing large areas to be surveyed and monitored; however, they have limitations. Green roofs can only be identified by this method if they are visible in the imagery, and small areas of vegetation (that is, not visible at 25 cm pixel size) are necessarily left out. Hand labelling also has limitations; there are edge cases where it is difficult to for a human to determine visually from the imagery whether a building has a green roof or where the edge of the green roof is due to shadows or poor condition of the green roof.

We compared performance between two sets of imagery collected in different years by the same company. However, these two sets of imagery are quite similar as they were collected by the same company, both in summer, and presumably with the

same or similar instruments, it may be that the model would not generalise to a model from a completely different source of imagery e.g. from satellite observations. Summer collection of optical aerial imagery is preferred because a high solar angle means better light conditions. When a model trained on 2021 imagery is tested against 2019 imagery, recall holds up well but there is a substantial difference in precision. More broadly, generalisation to completely different imagery sets (for example satellite imagery) would be best achieved by including examples from those sets during training. The trained model would not be expected to perform well on a completely unseen source of imagery without further training, as a diversity of imagery sources was not present during training. While relatively high-resolution satellite-imagery is available covering most cities in the world, these are generally not as high quality as the aerial imagery available in London; therefore, the same method applied to other cities may yield worse performance.

We have not attempted to separate different types of green roof (e.g. intensive, extensive, roof gardens). While types of plant may be differentiated to some extent in aerial imagery, important features like depth of substrate cannot. Some green roofs may be in poor condition from lack of water, and there may be cases of fake turf or other imitation vegetation being detected as green roofs: both of these could be better identified using multi-spectral imagery.

Performance of the building-intersection step is reliant on the alignment of the buildings footprints with the imagery. The OS building footprints are very accurate, especially for identifying courtyards within building footprints. We found that alignment between other imagery sets, and with other building footprint sources, was not as reliable. However, OS maps are only available in the Great Britain, as opposed to OpenStreetMap which has a more global coverage.

### 4.3    Comparison to other estimates

Our estimate of green roof area in the CAZ in 2021 ($2.3 \times 10^5$ $m^2$) is higher than the LRW2019 estimates and the AMR estimate for 2013 and 2015, but lower than the AMR estimates for 2017. For Greater London, the identified area is higher than the 2016 and 2017 estimated areas from LRW2019. While individual-building data from previous studies are not available for comparison, local-authority district (LAD) level data are available from for 2017 from LRW2019 (Livingroofs Enterprises Ltd, 2019; European Federation of Green Roof and Green Wall Associations (EFB) and Livingroofs.org on behalf of the Greater London Authority, 2019) In Figure 10, we compare our results for 2019 with the estimates for each LAD in 2017 from LRW2019: the results are strongly correlated, but some LADs have quite different results. According to this, most LADs have gained some green roof between 2017, with a few losing some. Newham (Nwm) and Hillingdon (Hdn) appear to have gained the most green roofs between 2017 and 2019. Our estimate for Havering (Hvg) is close to zero, because the 2019 imagery does not cover Havering (see Figure 2). Where estimates are differ by a small amount it may be due to differences in methodology or errors rather than a real change.

Examining the GLA's geospatial data (which is only public for the CAZ) (Greater London Authority, 2014) and infographics (Livingroofs Enterprises Ltd, 2019), we see multiple instances of ground-level parks being incorrectly identified as green roofs (e.g. Finsbury Square in Islington, Figure A1). Making use of the building footprint data enables us to avoid such misclassifications. There is also disagreement for the Barbican Centre (Figure A2), of which the full area is counted as a roof by the GLA results: this is a difficult edge-case, as the OS building footprints do not include the full area of the complex as a

building. Over the CAZ, we find that 4% of the the area of the Greater London Authority (2014) dataset does not intersect with OS building footprints. It also appears that in the GLA's geospatial data, an area slightly larger than the vegetation is usually selected, which may be due to the resolution of the input data. This demonstrates the utility of ensuring the coincidence of identified green roof patches with building footprints.

Wu and Biljecki (2021) report that proportion of buildings by area which have a green roof is 41.6% in Zurich, 24.8% in Berlin, and 17.2% in New York (London was not included in their survey). Comparing this with the results in Tables 7, 8, 9, we see that the City of London ranks between Berlin and New York at 21.0%. This method of ranking is sensitive to the geographic area included in the calculation if the concentration of green roofs varies between districts within a city. Furthermore, given our interest in rooftop vegetation as a climate adaptation strategy, the actual amount of vegetation seems more relevant than the total area of the building.

## 4.4 Distribution of green roof areas

As shown in Table 7, although larger total areas of green roof are present in some LADs, the City of London is has the highest concentration of green roofs in London. Especially high concentrations of green roofs are also seen in in Tower Hamlets.

Distribution of green roofed buildings within LADs is heterogeneous (see Fig 7). When LSOAs stand out as having relatively high green roof coverage, it is often due to a single large building or a cluster of buildings with green roofs.

Despite having the highest green roof coverage out of the LADs, only 3.9% of the building footprint of the City of London is covered by green roofs. The City of London has very low amounts of green cover generally, so it is consistent with policy (e.g. (Greater London Authority, 2021b, Policy G5)) that green roofs would be adopted there. However, the LRW 2008 report (Design for London et al., 2008) found that 32% of roof area in the City of London could be suitable for retrofitting with green roofs, so the current status is a long way from that proposed. As the dataset identifies individual buildings, in future work we will explore what kinds of buildings, and what areas, have adopted green roofs. Given that the area of vegetation in the City of London is overall quite low, it is possible that existing green roof coverage is making a difference to the thermal environment: a possibility that we will explore in a urban climate modelling study enabled by this data.

## 4.5 Use of the dataset

The dataset provides far greater detail than is available publicly from previous work in London. Green roof polygons are provided for individual areas of green roof, and are identifiable for individual buildings. This will enable new insights into the distribution of green roofs in London which were not possible before. For example, using the building use classifications given by the UKBuildings dataset, we can calculate the distribution of green roofs between building uses. As shown by Figure 11, non-residential buildings make up most of the buildings with green roofs (56%), with around 1.2% of non-residential building footprint area covered by green roofs compared to 0.3% of residential buildings. While a large fraction of green roofs occur on residential buildings, only a small proportion of residential buildings have a green roof. This illustrates the utility that this level of detail brings. Future work will extend this analysis to look in detail at the characteristics of buildings that have green roofs in London.

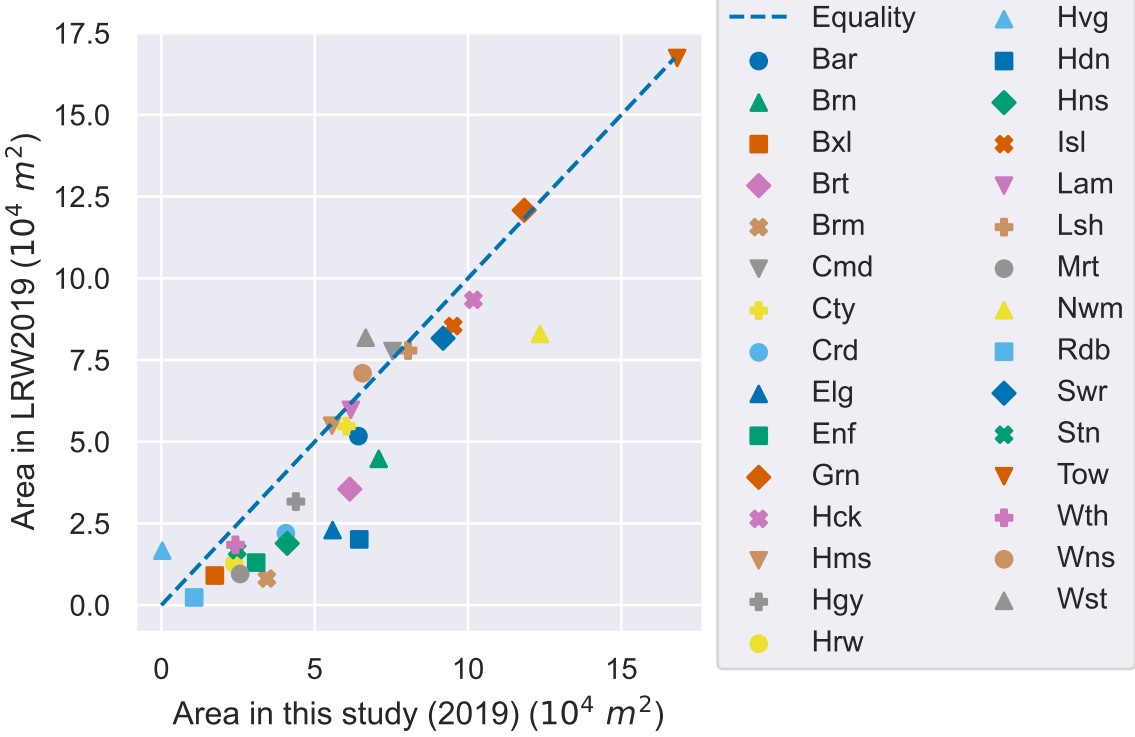

**Figure 10.** Scatter plot showing estimated green roof area in LADs of Greater London, comparing the estimates from Livingroofs Enterprises Ltd (2019) (2017) to our estimates (2019).

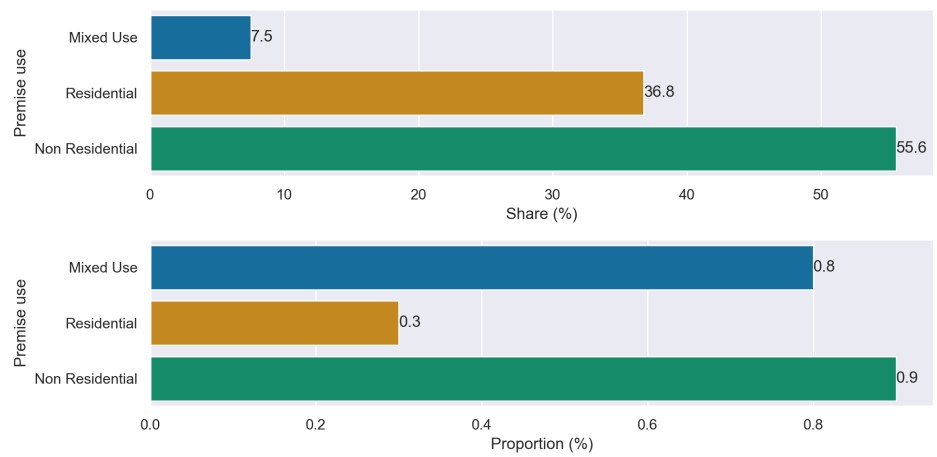

**Figure 11.** Share and proportion of green roofs for different building uses. Most green roofs are on non-residential buildings. Mixed use refers to buildings comprising both residential and non-residential uses.

## 5 Conclusions

In this study, we produced a survey of green roofs in London using automatic segmentation of aerial imagery. The segmentation model shows strong generalisation to unseen imagery. The resulting geospatial dataset is made available for further research. We identified areas which have a high prevalence of green roofs; especially the City of London and parts of Tower Hamlets. We highlighted some of the difficulties of producing such a dataset: especially that low prevalence of green roofs means that the classification problem is highly imbalanced, which can create problems for machine-learning algorithms. Furthermore, we demonstrate the importance of excluding ground-level vegetation from surveys of green roofs by ensuring areas classified as green roofs are coincident with building footprints.

This dataset covers the whole of Greater London and provides data at the single-building level, which other publicly available datasets cannot. We demonstrated how the resulting dataset can be used to extract information about the distribution of green roofs between districts, as well as using single-building-level data to cross reference green roof coverage with building use. In future work, we will use this geospatial dataset to further explore the characteristics and uses of buildings and neighbourhoods which have green roofs as well as those with potential for more green infrastructure, and to quantify the thermal effects of green roofs on London's micro-climate through urban climate modelling.

## 6 Code and data availability

Code and data generated by this project are available for download 10.5281/zenodo.7603123 (Simpson et al., 2023). The geospatial data is stored in GeoJSON format, and can be read with GIS applications such as QGIS, ArcMap, or Fiona.

Aerial imagery was used under license from GetMapping Plc. Ordnance Survey data was used under license. These licensed data are available under educational license https://digimap.edina.ac.uk.

*Author contributions.* All authors participated in the conceptualisation of the paper. CS did the investigation and wrote the initial draft, with all others participating in review and editing.

*Competing interests.* The authors declare no competing interests.

*Acknowledgements.* CS, OB, and CH are supported by the Wellcome HEROIC Project (216035/Z/19/Z). CH is also supported by a NERC fellowship (NE/R01440X/1). The authors acknowledge the use of the UCL Myriad High Performance Computing Facility (Myriad@UCL), and associated support services, in the completion of this work. Thanks to Zaid Chalabi, Lora Fleming, James Grellier, Jon Taylor and Tim Taylor who all provided feedback on an early draft.

**Table A1.** Table listing the hyperparameters that were tuned, which values were tested, and the final value used for classification.

| Parameter | Tested values | Final value |
|---|---|---|
| Loss function | Cross entropy, Lovasz, Focal, F-measure | F-measure |
| Learning rate | 5.e-3, 5.e-4, 5.e-5, 5.e-6 | 5.e-5 |
| Random augmentations | None; flips and 90° rotations; crops and flips and 90 ° rotations; flips and fully random rotations; 90% crops and flips and 90° rotations; flips and 90° rotations and sharpness; flips and 90° rotations and sharpness; elastic distortion; alterations to gamma and colour | Flips, rotations, elastic distortions, alterations to gamma and colour |
| Max epochs | 100 | 100 |
| Pretrained model frozen | True, False | True |

**Table A2.** Grid references of hand-labelled areas. Grid references are in the OSGB 1936 system.

| | | | | | |
|---|---|---|---|---|---|
| TQ2280 | TQ2474 | TQ2481 | TQ2678 | TQ2781 | TQ2784 |
| TQ2872 | TQ3073 | TQ3176 | TQ3181 | TQ3184 | TQ3280 |
| TQ3281 | TQ3282 | TQ3290 | TQ3373 | TQ3478 | TQ3486 |
| TQ3681 | TQ3775 | TQ3783 | TQ3872 | TQ4084 | TQ4180 |
| TQ4283 | | | | | |

**Table A3.** Full confusion matrix for the green roof identification method, calculated based on area. TP, TN, FP, FN are as a proportion of total building footprint area in the hand-labelled areas.

| Dataset | K-fold | Land area ($km^2$) | Built area ($km^2$) | TP | TN | FP | FN |
|---|---|---|---|---|---|---|---|
| Testing 2021 | 1 | 3.9 | 1.282 | 0.0050 | 0.9879 | 0.0024 | 0.0048 |
| | 2 | 3.9 | 1.282 | 0.0047 | 0.9876 | 0.0026 | 0.0051 |
| | 3 | 3.9 | 1.282 | 0.0047 | 0.9872 | 0.0030 | 0.0051 |
| | 4 | 3.9 | 1.282 | 0.0054 | 0.9876 | 0.0026 | 0.0044 |
| | average | 3.9 | 1.282 | 0.0049 | 0.9876 | 0.0027 | 0.0048 |
| Testing 2019 | 1 | 3.9 | 1.282 | 0.0047 | 0.9877 | 0.0043 | 0.0034 |
| | 2 | 3.9 | 1.282 | 0.0041 | 0.9878 | 0.0042 | 0.0040 |
| | 3 | 3.9 | 1.282 | 0.0048 | 0.9861 | 0.0058 | 0.0033 |
| | 4 | 3.9 | 1.282 | 0.0052 | 0.9880 | 0.0039 | 0.0029 |
| | average | 3.9 | 1.282 | 0.0047 | 0.9874 | 0.0045 | 0.0034 |
| Validation | 1 | 4.0 | 1.333 | 0.0060 | 0.9887 | 0.0021 | 0.0033 |
| | 2 | 4.0 | 1.278 | 0.0070 | 0.9851 | 0.0032 | 0.0047 |
| | 3 | 3.9 | 1.255 | 0.0060 | 0.9839 | 0.0030 | 0.0071 |
| | 4 | 4.0 | 1.261 | 0.0079 | 0.9844 | 0.0026 | 0.0052 |
| | average | 4.0 | 1.282 | 0.0067 | 0.9856 | 0.0027 | 0.0050 |
| Training | 1 | 12.0 | 3.798 | 0.0089 | 0.9851 | 0.0029 | 0.0031 |
| | 2 | 12.0 | 3.853 | 0.0075 | 0.9863 | 0.0025 | 0.0036 |
| | 3 | 12.1 | 3.875 | 0.0077 | 0.9860 | 0.0033 | 0.0030 |
| | 4 | 12.0 | 3.870 | 0.0084 | 0.9863 | 0.0028 | 0.0025 |
| | average | 12.0 | 3.849 | 0.0081 | 0.9859 | 0.0029 | 0.0031 |

**Table A4.** Full performance metrics for the green roof identification method, calculated based on area.

| Dataset | K-fold | Accuracy | IoU | Precision | Recall | F-score |
|---|---|---|---|---|---|---|
| Testing 2021 | 1 | 0.9929 | 0.4134 | 0.6801 | 0.5131 | 0.5849 |
| | 2 | 0.9923 | 0.3774 | 0.6423 | 0.4778 | 0.5480 |
| | 3 | 0.9919 | 0.3659 | 0.6075 | 0.4792 | 0.5358 |
| | 4 | 0.9930 | 0.4325 | 0.6720 | 0.5482 | 0.6038 |
| | average | 0.9925 | 0.3970 | 0.6505 | 0.5046 | 0.5683 |
| Testing 2019 | 1 | 0.9923 | 0.3782 | 0.5216 | 0.5790 | 0.5488 |
| | 2 | 0.9918 | 0.3329 | 0.4946 | 0.5044 | 0.4995 |
| | 3 | 0.9909 | 0.3436 | 0.4509 | 0.5907 | 0.5114 |
| | 4 | 0.9932 | 0.4328 | 0.5697 | 0.6431 | 0.6042 |
| | average | 0.9921 | 0.3707 | 0.5072 | 0.5793 | 0.5408 |
| Validation | 1 | 0.9947 | 0.5298 | 0.7454 | 0.6468 | 0.6926 |
| | 2 | 0.9922 | 0.4735 | 0.6893 | 0.6020 | 0.6427 |
| | 3 | 0.9899 | 0.3754 | 0.6696 | 0.4607 | 0.5459 |
| | 4 | 0.9923 | 0.5048 | 0.7538 | 0.6045 | 0.6709 |
| | average | 0.9923 | 0.4666 | 0.7148 | 0.5733 | 0.6363 |
| Training | 1 | 0.9940 | 0.5957 | 0.7514 | 0.7419 | 0.7466 |
| | 2 | 0.9938 | 0.5497 | 0.7487 | 0.6741 | 0.7094 |
| | 3 | 0.9937 | 0.5510 | 0.7018 | 0.7194 | 0.7105 |
| | 4 | 0.9947 | 0.6120 | 0.7479 | 0.7710 | 0.7593 |
| | average | 0.9941 | 0.5773 | 0.7374 | 0.7266 | 0.7320 |

**Table A5.** Full confusion matrix for the green roof identification method, calculated based on counts of buildings. TP, TN, FP, FN are as a proportion of total building footprint area.

| Dataset | K-fold | Building count | TP | TN | FP | FN |
|---|---|---|---|---|---|---|
| Testing 2021 | 1 | 12026 | 0.0063 | 0.9835 | 0.0075 | 0.0027 |
| | 2 | 12026 | 0.0067 | 0.9742 | 0.0168 | 0.0022 |
| | 3 | 12026 | 0.0068 | 0.9727 | 0.0183 | 0.0022 |
| | 4 | 12026 | 0.0070 | 0.9791 | 0.0119 | 0.0020 |
| | average | 12026 | 0.0067 | 0.9774 | 0.0136 | 0.0023 |
| Testing 2019 | 1 | 12026 | 0.0067 | 0.9739 | 0.0181 | 0.0013 |
| | 2 | 12026 | 0.0064 | 0.9617 | 0.0304 | 0.0016 |
| | 3 | 12026 | 0.0067 | 0.9543 | 0.0378 | 0.0012 |
| | 4 | 12026 | 0.0068 | 0.9618 | 0.0302 | 0.0012 |
| | average | 12026 | 0.0067 | 0.9629 | 0.0291 | 0.0013 |
| Validation | 1 | 11505 | 0.0077 | 0.9832 | 0.0066 | 0.0024 |
| | 2 | 11724 | 0.0078 | 0.9701 | 0.0195 | 0.0026 |
| | 3 | 11167 | 0.0081 | 0.9684 | 0.0215 | 0.0021 |
| | 4 | 11820 | 0.0085 | 0.9767 | 0.0132 | 0.0016 |
| | average | 11554 | 0.0080 | 0.9746 | 0.0152 | 0.0022 |
| Training | 1 | 30932 | 0.0065 | 0.9820 | 0.0100 | 0.0016 |
| | 2 | 30717 | 0.0072 | 0.9705 | 0.0209 | 0.0015 |
| | 3 | 31069 | 0.0070 | 0.9674 | 0.0242 | 0.0014 |
| | 4 | 30731 | 0.0071 | 0.9758 | 0.0159 | 0.0012 |
| | average | 30862 | 0.0069 | 0.9739 | 0.0177 | 0.0014 |

**Table A6.** Full Performance metrics for the green roof identification method, calculated based on building counts.

| Dataset | K-fold | Accuracy | IoU | Precision | Recall | F-score |
|---|---|---|---|---|---|---|
| Testing 2021 | 1 | 0.9932 | 0.4266 | 0.6354 | 0.5648 | 0.5980 |
| | 2 | 0.9919 | 0.3938 | 0.5478 | 0.5833 | 0.5650 |
| | 3 | 0.9916 | 0.3952 | 0.5280 | 0.6111 | 0.5665 |
| | 4 | 0.9928 | 0.4387 | 0.5913 | 0.6296 | 0.6099 |
| | average | 0.9924 | 0.4128 | 0.5721 | 0.5972 | 0.5844 |
| Testing 2019 | 1 | 0.9919 | 0.3718 | 0.4915 | 0.6042 | 0.5421 |
| | 2 | 0.9899 | 0.3260 | 0.4097 | 0.6146 | 0.4917 |
| | 3 | 0.9854 | 0.2798 | 0.3163 | 0.7083 | 0.4373 |
| | 4 | 0.9896 | 0.3455 | 0.4099 | 0.6875 | 0.5136 |
| | average | 0.9892 | 0.3256 | 0.3934 | 0.6536 | 0.4912 |
| Validation | 1 | 0.9937 | 0.5000 | 0.7157 | 0.6239 | 0.6667 |
| | 2 | 0.9902 | 0.3883 | 0.5252 | 0.5984 | 0.5594 |
| | 3 | 0.9909 | 0.4171 | 0.5407 | 0.6460 | 0.5887 |
| | 4 | 0.9934 | 0.5185 | 0.6614 | 0.7059 | 0.6829 |
| | average | 0.9920 | 0.4516 | 0.6024 | 0.6433 | 0.6222 |
| Training | 1 | 0.9937 | 0.4771 | 0.5920 | 0.7108 | 0.6460 |
| | 2 | 0.9923 | 0.4381 | 0.5428 | 0.6943 | 0.6093 |
| | 3 | 0.9914 | 0.4145 | 0.4922 | 0.7241 | 0.5860 |
| | 4 | 0.9927 | 0.4548 | 0.5471 | 0.7294 | 0.6252 |
| | average | 0.9925 | 0.4444 | 0.5404 | 0.7146 | 0.6154 |

**Table A7.** Standard deviation of performance metrics between folds, calculated using area.

| Dataset | Accuracy | IoU | Precision | Recall | F-score |
|---|---|---|---|---|---|
| Testing 2021 | 0.0005 | 0.0310 | 0.0330 | 0.0333 | 0.0317 |
| Testing 2019 | 0.0010 | 0.0450 | 0.0497 | 0.0572 | 0.0471 |
| Validation | 0.0019 | 0.0677 | 0.0414 | 0.0812 | 0.0647 |
| Training | 0.0004 | 0.0316 | 0.0238 | 0.0409 | 0.0254 |

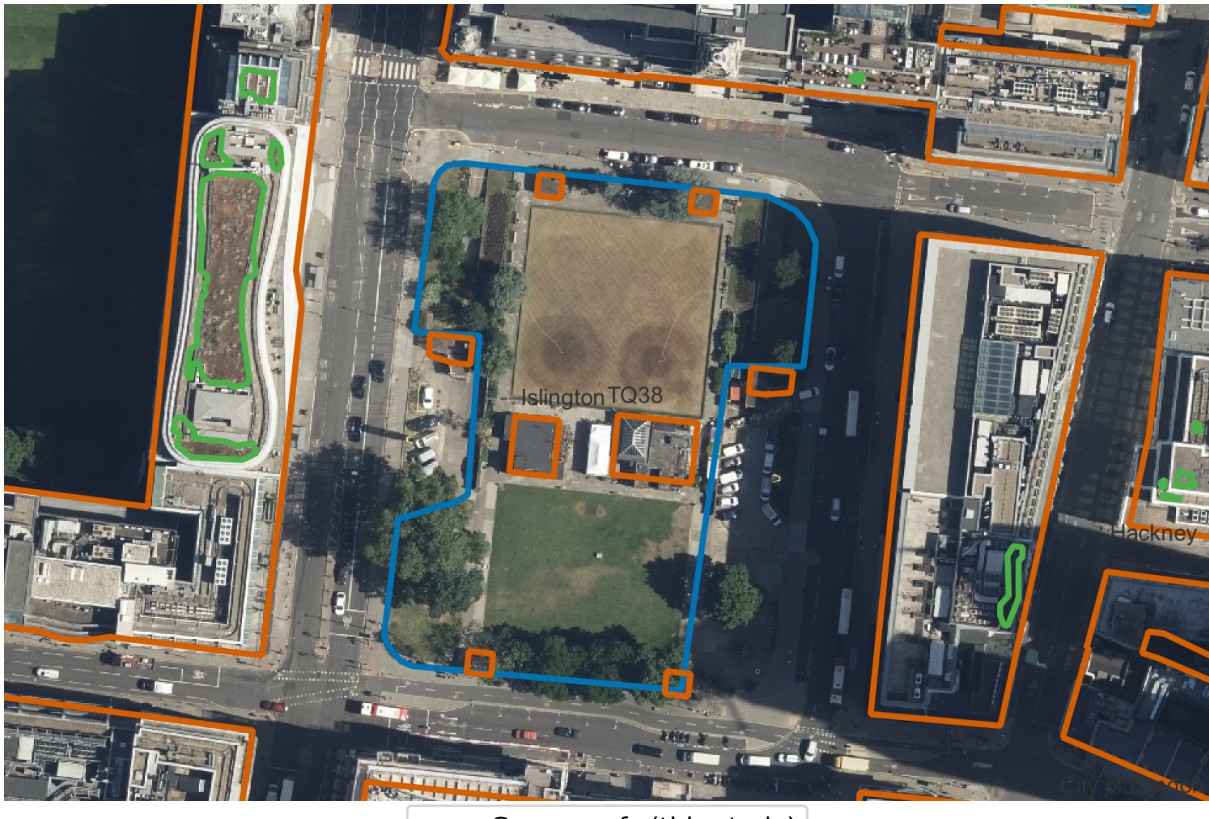

**Figure A1.** Example of ground-level green space misclassified as green roof in GLA dataset (in blue) (Greater London Authority, 2014). Building outlines according to OS VML are shown in orange, our results are shown in green. Image location is shown in Figure A3. Imagery ©GetMapping Plc. Building polygons are OS data ©Crown copyright and database rights 2022.

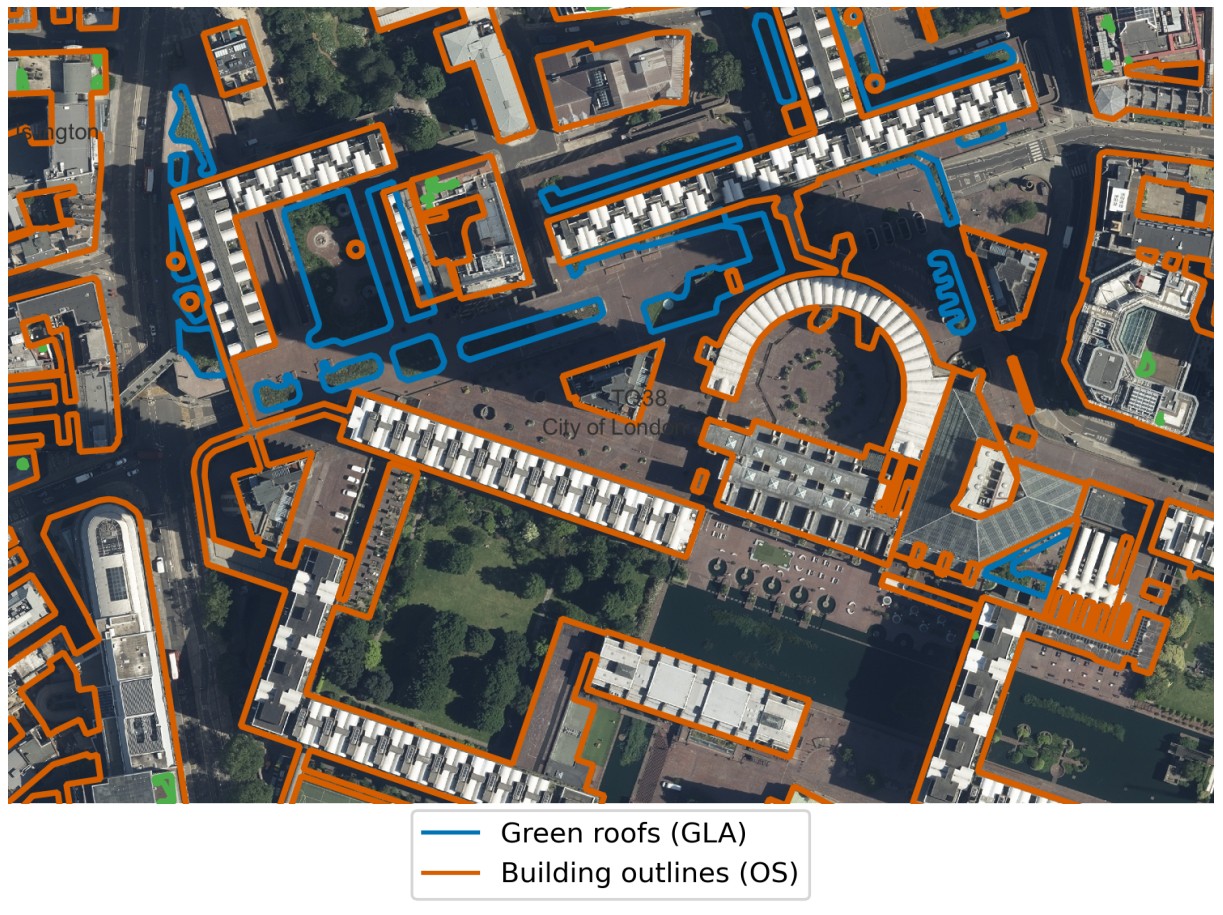

**Figure A2.** Example of disagreement between our result and GLA dataset (Greater London Authority, 2014) (in blue) due to building outlines. Building outlines according to OS VML are shown in orange. The green areas of the Barbican Centre are excluded in our analysis, as the OS VML does not identify them as within a building footprint. Image location is shown in Figure A3. Imagery ©GetMapping Plc. Building polygons are OS data ©Crown copyright and database rights 2022.

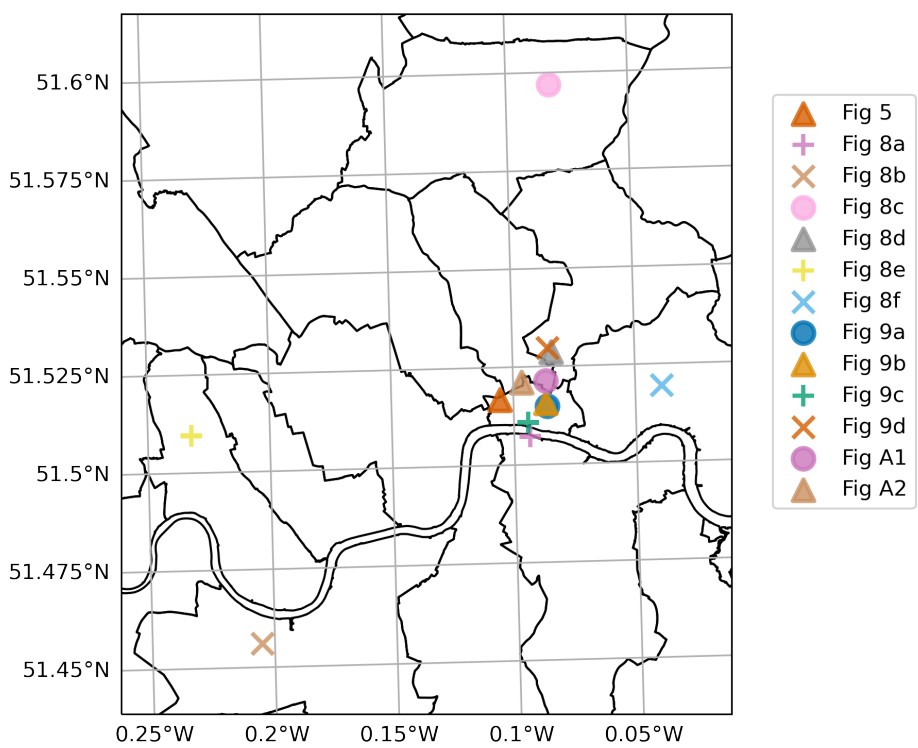

**Figure A3.** Locations of images in this paper.

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
