# Peer review of "An Open-Source Automatic Survey of Green Roofs in London using Segmentation of Aerial Imagery"

_Earth System Science Data, 2022_

## Author Comment (AC1)

**Manuscript essd-2022-259, First Revision**

Charles H. Simpson      Oscar Brousse      Nahid Mohajeri      Michael Davies

Clare Heaviside

September 23, 2022

We thank the editors and reviewers for their helpful comments and appreciation. Please find point-by-point responses below. **Our responses look like this**, *quotes from the changed text look like this*.

Please note that RC2 and RC3 are the same comment posted twice, so we only respond once.

**1 RC1**

**1.1 General comments**

This is a detailed, well-written manuscript that distinctly describes the new vectorised green roofs dataset and the deep-learning method applied to automatically detect green roofs in visible aerial imagery. It is easy to gain access to the dataset and supporting code. The methods section provides useful information pertaining to hyperparameter tuning which is important for reproducibility. The results provide valuable insight into relative green roof coverage across London boroughs (e.g., Figures 4 and 5). Although this information is already contained within the datasets referenced and compared to within the manuscript (LRW2019 and London Plan AMR 16), it is evident this is the first time this data has been provided in an open access format.

**Thank you for supporting and taking the time to review our paper.**

My comments are primarily minor or typographical in nature, apart from one major concern. This relates to the lack of evidence that the trained U-Net can suitably generalise to other locations or other imagery of London. One of the benefits of training a neural network is the ability to apply the tool to automatically detect the feature(s) of interest in new images. Therefore, it is a concern if it is not possible to do this. The trained U-Net was not tested on imagery captured during different years or seasons in the year when lighting conditions may alter the appearance of the green roofs in the imagery. It is highlighted within the discussion section that the trained U-Net produced a lot of false positive results (over-predicted green roof coverage) in some Eastern boroughs of London. This is attributed to the potential use of a different collection instrument, highlighting that the trained U-Net may not be able to generalise to imagery captured using different sensors. It is suggested that the U-Net is applied to imagery captured during different years to test the model's ability to generalise. The imagery does not nearly need to cover the whole of London but cover one or two study locations to demonstrate the ability or otherwise for the trained U-Net to detect green roofs in a variety of settings.

**We would not expect the model to generalise well to different collection dates or instruments, as it has not be exposed to diverse collection dates or instruments during training. This does not affect the validity of the dataset we are presenting. We have added the following to the text to clarify this in sec 4.2.**

*The majority of the Greater London region is covered by a single acquisition date (2019-06-29, see Figure 2), and this also contains the majority of the data included in training. Summer collection of optical aerial imagery is preferred because a high solar angle means better light conditions. We found that there was a higher rate of false positives east of OSGB37 Easting $5.5{\times}10^5$ m, for which imagery was acquired on 2018-08-02 and 2018-09-01 (see Figure 2). Although part of the north of the region is covered by another acquisition date (2019-08-22), some data from this acquisition date was included in training, as can be seen by comparing Figures 2 and 4, and generally classification performance is not visibly different in this area. Performance was good for acquisition dates which were included in training. As a result, data in the 2018-08-02 and 2018-09-01 area were excluded from the dataset, affecting parts of Bromley, Havering, Bexley, and Barking and Dagenham. The excluded area contains a large amount of agricultural land, woodland, surburban streets, and industrial areas which typically do not have green roofs, so few if any green roofs are missed.*

*Generalisation to different imagery sets is achievable provided that both positive and negative examples from each imagery dataset are included. The most northern training area, TQ3290 was included specifically to improve performance in the northern area covered by the collection date 2019-08-22. This is not viable for the eastern areas covered by 2018-08-02 and 2018-09-01 because we could not find any positive examples in this area with which to train. This means that updating this dataset in future using new imagery may require further training of the model on the new imagery.*

*More broadly, generalisation to completely different imagery sets (for example satellite imagery) would be best achieved by including examples from those sets during training. The trained model would not be expected to perform well on a completely unseen source of imagery without further training, as a diversity of imagery sources was not present during training. While relatively high-resolution satellite-imagery is available covering most cities in the world, these are generally not as high quality as the aerial imagery available in London; therefore, the same method applied to other cities may yield worse performance.*

[Figure]

Figure 2: Collection dates for the aerial imagery mosaic covering Greater London.

[Figure]

Figure 4: Map of hand labelled areas.

**1.2   Specific comments**

Line 105- Please detail the number of images mosaiced to cover the whole London study area and the time period covering the first and last image. Also, is there information on the time of day when the images were captured?

**We have added a figure showing the collection dates of the mosaic (now Fig 2, see above). Time of day when the images were captured is not available to us.**

Section 2.3- Was any pre-processing conducted e.g., pixel value normalisation?

**We have added this to the text in sec 2.5: *Pixel values were normalized to match ImageNet during training and prediction.***

Section 2.3- Please provide a little more information on how the trained U-Net was applied to unseen images. E.g., were images tiled or patched prior to input? If so, how were outputs mosaiced and were there any issues with predictions at the edge of the tiles? Section 2.3- How was it defined whether

a pixel was positive or negative? Was a threshold value applied to the U-Net outputs and was a consistent value applied across images/ image patches?

**We have added the following to sec 2.6:** *Predicted segmentation masks were generated from the trained model using the same tiling method as used for training. The same prediction probability threshold of 0.5 was applied across the whole domain.*

*In previous work, morphological opening and closing have been used on the classification masks as a post-processing step: these are filters that remove small isolated positive areas and fill in small negative areas respectively. This can be useful for filling in small gaps resulting from the tiling of the input imagery. We tested these methods with our own models and imagery, but found that morphological opening of the classification masks increased recall but decreased precision, overall decreasing F-score; whereas, morphological closing did not have any substantial effect on F-score. Therefore, we decided not to include these post-processing steps in our final classification pipeline.*

*From the binary masks produced by the segmentation algorithm, we extracted green roof candidate polygons. Single pixel-wide gaps are visible in some of the candidate polygons as a result of the tiling of the input images and not using morphological closing. The intersection was then taken between the candidate polygons and the OS VML building footprints, to remove any candidate polygons that did not intersect with a building footprint. This process helped to reduce the false positive rate because the segmentation algorithm can incorrectly identify ground-level green cover as a green roof. The post-processed segmentation results were spatially joined with the UKBuildings layer in order to identify which individual buildings have green roofs, and so calculate the number of buildings covered.*

Line 169- was class imbalance considered within the loss functions? If so, how?

**We have added the following sentence in sec 2.5** *The loss functions were weighted by the frequency of the classes to account for class imbalance.*

Figure A2- The orange and blue labels are the wrong way round in the legend and image description. Figure A3- The orange and blue labels are the wrong way round in the legend and image description. There are also no green lines in the image.

**Thanks, we have corrected the mistakes with the legends.**

**1.3 Technical corrections**

Line 86- change 'an' to 'a'

115- missing word between 'hand-labelled' and 'are'. Should it be polygons/ datapoints?

152- 'm' should be italics.

Throughout- be consistent in use of 'hyperparameter' or 'hyper-parameter'.

**We have corrected the above typographical errors.**

Figure 6- please add in small-scale schematic of London to show where the example images are from.

**We have added a supplementary figure A4 showing the locations of the example images.**

[Figure]

Figure A04: Locations of images in this paper.

**2   RC2 / RC3**

This is a well written paper that develops a deep learning-based green roofs' mapping framework in incorporating very high-resolution remote sensing satellites. Also, this study used data augmentation to improve the accuracy of the mapping and provide a robust dataset of green roof locations and areas in London. Overall, it is helpful for others to study aerial remote sensing mapping. However, there are some problems, which must be solved before it is considered for publication. If the following problems are well-addressed, this reviewer believe that the essential contribution of this paper are important for designing sustainable buildings and studying urban microclimates.

**Thank you for supporting and taking the time to review our paper.**

**2.1   General Comments**

In ABSTRACT: authors are suggested to refine the abstract, focusing on the novelty of the research rather than providing extensive background information. In addition, semantic segmentation models coupled with data augmentation strategies are important in this study. It is helpful to provide the validation accuracy accordingly in this section.

**We have edited the abstract to shorten the background information and mention segmentation earlier. We have put the accuracy, precision, and recall for the test set in the abstract. Please see the new abstract:**

*Green roofs can mitigate heat, increase biodiversity, and attenuate storm water, giving some of the benefits of natural vegetation in an urban context where ground space is scarce. To guide the design of more sustainable and climate resilient buildings and neighbourhoods, there is a need to assess the existing status of green roof coverage and explore the potential for future implementation. Therefore, accurate information on the prevalence and characteristics of existing green roofs, but this information is currently lacking. Segmentation algorithms have been used widely to identify buildings and land cover in aerial imagery. Using a machine-learning algorithm based on U-Net to segment aerial imagery, we surveyed the area and coverage of green roofs in London, producing a geospatial dataset (Simpson et al., 2022). We estimate that there was $0.19\ km^2$ of green roof in the Central Activities Zone (CAZ) of London, ($0.81\ km^2$) in Inner London, and ($1.25\ km^2$) in Greater London in the year 2019. This corresponds to 1.6% of the total building footprint area in the CAZ, and 1.0% in Inner London. There is a relatively higher concentration of green roofs in the City of London (the historic financial district), covering 3.1% of the total building footprint area. Test set accuracy was 0.996, with an f-score of 0.757. We improve on previous studies by including more negative examples in the training data, by experimenting with different data augmentation methods, and*

*by requiring coincidence between vector building footprints and green roof patches. The survey covers 1463~$km^2$ of Greater London, making this the largest open automatic survey of green roofs in any city. This dataset will enable future work exploring the potential of green roofs in London and on urban climate modelling.*

In INTRODUCTION: authors are suggested to start broad in the general background, then narrow in on the relevant topic that will be pursued in the paper. The introduction sections are to highlight the challenges currently faced by green roofs' mapping research. I suggest that the first three paragraphs be summarized in one paragraph. In addition, the detailed description of the mentioned algorithms (UNet) can be moved to the second section since we are not developing new models.

**We have reduced the general background information. We have moved some of the detail about the U-Net to the Methods section.**

In DATA and METHODS: there are many datasets mentioned and may be clearer if summarized in a table. For example, recording information such as the coverage of the study area, the date of data acquisition and the spatial resolution of the imagery. We all know that convolutional neural networks are data-driven models. Well, how many positives and negatives are there before and after using data augmentation, and is there an improvement in model performance and by how much? Also, although deep learning or semantic segmentation models are a black box, it is helpful to provide formulas. Besides, I have some doubts about the details of the algorithm framework. For example, the loss curve during optimization. Furthermore, the general area of the study area is larger than a 256x256 image patches. How did you deal with it when predicting?

**We have added the date of acquisition for the imagery as a figure (now Fig 2).**

[Figure]

Figure 2: Collection dates for the aerial imagery mosaic covering Greater London.

Our data augmentation method does not alter the number of positives and negative tiles in the training dataset. Changes are applied to tiles randomly during each training epoch. We have clarified this in sec 2.5.

*Augmentation was applied randomly and independently each training epoch, with equal probability to positive and negative tiles.*

We have added a Table summarising the datasets as requested.

We have noted the difference in f-score between the best performing augmentation and the base case in sec 3.1. *The best performing augmentation had a higher f-score by 0.146 than the base case.*

We have not introduced any new mathematical formulae in this study, it is not clear which formulae the reviewer wants to be included.

Datasets described in this section are summarised in Table 2.

| Dataset | Type | Source | Explanation |
|---|---|---|---|
| Imagery | RGB raster | Getmapping Inc. | Cloud free vertical aerial imagery mosaic. |
| OS VML | Vector | Ordance Survey | Large scale building outlines. |
| UKBuildings | Vector | Verisk Ltd. | Building footprints. |
| LSOA outlines | Vector | Office of National Statistics | Lower-super output areas from 2011 Census. |
| LAD outlines | Vector | Office of National Statistics | Local authority districts boundaries. |

**Table 2.** Input geospatial dataset summary.

Figure 1: Table 2

**Prediction is tiled in the same way as during training - we have added a sentence that says this sec 2.6. *Predicted segmentation masks were generated from the trained model using the same tiling method as used for training.***

In RESULTS: several tables record the model performance. Recommendations for the structure of research paper. In Tables 3 and 5, I found that the values of TP, TN, FP, and FN vary greatly. Also, despite the high accuracy of the models in Tables 4 and 6, the performance is weaker on the IOU metrics. Why is this happening?

**The difference between Tables 3 and 5 (now Table 5 and 7) is that the former reports values calculated as area, whereas the latter uses counts of buildings. Recall is higher and precision lower based on building counts, meaning that a patch of green roof anywhere on the building is counted as a "hit" so both false positives and true positives are more common. We have clarified this in sec 4.1.**

*Counting buildings rather than measuring areas increases the number of positives (both false positives and true positives) as an example is counted as positive if any part of the building is identified as positive: this leads to higher recall and lower precision in Table 7 compared to Table 5.*

The image segmentation algorithm used is supervised classification. As a result, the classification results are constrained by the labels, and the model's generalization is limited. Even though we are able to assign labeling tasks to individuals, we cannot classify locations where the image is occluded. Is the current weakly supervised or unsupervised segmentation a significant advancement?

**No algorithm will be able to classify locations where the image is occluded. This study uses a supervised method, we did not test any unsupervised or weakly supervised methods.**

**3   RC4**

Summary: In the study titled "An Open-Source Automatic Survey of Green Roofs in London using Segmentation of Aerial Imagery", the authors use a convolutional neural network to detect green roofs over buildings in London from aerial imagery. The paper is generally well-written and the authors sufficiently describe the model architecture, hyperparameter tuning, and some of the uncertainties in the results. The dataset will be useful for future studies on the urban climate of London and for informing heat mitigation strategies. I do have a few questions and concerns about the methodology, particularly its generalizability, that should be addressed before the paper can be considered for publication.

**Thank you for supporting and taking the time to review our paper.**

**3.1 Major Comments**

I am a bit confused by how the tiles were split. The authors mention that every split contains both positive and negative classes. Is there any threshold used for what fraction can be positive or negative? Or can it, in theory, be a single negative class and the rest being positive? Additionally, was only one split used for the model training, validation, and testing. This is important since different random splits of training data might produce different results. Did the authors check for consistency across random training sets?

**We have added the following to sec 2.5**

*We refer to areas labelled with no green roof as negative, and those labelled with green roof as positive. All tiles within the hand-labelled areas were used. In order ensure that batches would contain positive examples, we over-sampled positive tiles by repetition during training so that they were equally prevalent as the fully negative tiles. Tiles were split randomly into training (80%), validation (10%), and testing sets (10%). The random split was performed separately for positive and fully negative tiles to ensure all splits contained both classes. For the purpose of this split a tile was positive if any pixel within it was positive, and negative otherwise.*

**We did not perform cross validation across random training sets.**

Aerial imagery for summer 2019 was used for the analysis. Was this a single image or multiple images mosaiced to cover the whole area? What were the dates of acquisition of these images? How was the presence of clouds in these images accounted for? There is not enough metadata about the imagery to understand the baseline observations.

**The imagery is a cloud free mosaic. We have clarified this in sec 2.1 and added the dates of acquisition in figure 2.**

*The imagery used for segmentation was colour (red, green, blue) raster images from a cloud-free mosaic of aerial imagery at 25 cm horizontal resolution.*

[Figure]

Figure 2: Collection dates for the aerial imagery mosaic covering Greater London.

Related to the previous point, it is unclear how generalizable these results and the U-net are. Would one expect the results to be generally replicable using an imagery for winter? Or if we use observations from a different satellite with the same spatial resolution?

**We added the following to the text in sec 4.2 to address generalisation.**

*The majority of the Greater London region is covered by a single acquisition date (2019-06-29, see Figure 2), and this also contains the majority of the data included in training. Summer collection of optical aerial imagery is preferred because a high solar angle means better light conditions. We found that there was a higher rate of false positives east of OSGB37 Easting $5.5 \times 10^5$ m, for which imagery was acquired on 2018-08-02 and 2018-09-01 (see Figure 2). Although part of the north of the region is covered by another acquisition date (2019-08-22), some data from this acquisition date was included in training, as can be seen by comparing Figures 2 and 4, and generally classification*

*performance is not visibly different in this area. Performance was good for acquisition dates which were included in training. As a result, data in the 2018-08-02 and 2018-09-01 area were excluded from the dataset, affecting parts of Bromley, Havering, Bexley, and Barking and Dagenham. The excluded area contains a large amount of agricultural land, woodland, surburban streets, and industrial areas which typically do not have green roofs, so few if any green roofs are missed.*

*Generalisation to different imagery sets is achievable provided that both positive and negative examples from each imagery dataset are included. The most northern training area, TQ3290 was included specifically to improve performance in the northern area covered by the collection date 2019-08-22. This is not viable for the eastern areas covered by 2018-08-02 and 2018-09-01 because we could not find any positive examples in this area with which to train. This means that updating this dataset in future using new imagery may require further training of the model on the new imagery.*

*More broadly, generalisation to completely different imagery sets (for example satellite imagery) would be best achieved by including examples from those sets during training. The trained model would not be expected to perform well on a completely unseen source of imagery without further training, as a diversity of imagery sources was not present during training. While relatively high-resolution satellite-imagery is available covering most cities in the world, these are generally not as high quality as the aerial imagery available in London; therefore, the same method applied to other cities may yield worse performance.*

[Figure]

Figure 4: Map of hand labelled areas.

**3.2   Minor comments**

A constraint on accuracy of the final dataset is the OS footprint data. The authors mention that it is "very accurate" in the Limitations section. It would be good to have a quantitative estimate here. Are the OS data also for 2019?

**We did not make a quantitative estimate of the accuracy of the OS footprint data, and we think to do so would be a large and complicated task that would not give much benefit to our study. To be clear, we do not think that the OS footprint data is a major source of uncertainty in this study, but we think the availability of high-quality building footprint data could be a constraint if the method were applied in another country. We simply compared the two available datasets (UKBuildings and OS VML) to the aerial imagery in the locations we were labelling, and examined them visually. We were able to see that while in general both were well aligned, there were cases where the UKBuildings footprints included courtyards as part of a building. The OS data are dated April 2019, we have now included this in the text.**

Table 1: Since this includes two different survey areas, would be good to also add the green roof areas as fraction of total area or total building footprint area for easy comparison. **We have added the fraction of total building footprint area to this table.**

**Table 1.** Previous estimates of green roof area in London. CAZ refers to a central area of London, see Subsection 2.1. Data from (European Federation of Green Roof and Green Wall Associations (EFB) and Livingroofs.org on behalf of the Greater London Authority, 2019) and (Greater London Authority, 2021a).

| Survey area | Source | Survey year | Green roof area ($10^5\ m^2$) | % of built area |
|---|---|---|---|---|
| CAZ | London Plan AMR 16 | 2013 | 1.75 | 1.5 |
| CAZ | London Plan AMR 16 | 2015 | 2.2 | 1.9 |
| CAZ | London Plan AMR 16 | 2017 | over 2.9 | 2.5 |
| CAZ | LRW2019 | 2016 | 1.5 | 1.3 |
| CAZ | LRW2019 | 2017 | 2.1 | 1.8 |
| Greater London | LRW2019 | 2016 | 11.0 | 0.43 |
| Greater London | LRW2019 | 2017 | 15.0 | 0.59 |

Figure A1: Please add the r2 value and equation of the line of best fit here for context

**There is no line of best fit shown in this figure, the line shown is y=x, labelled "Equality" in the legend.**

In the discussion, the authors talk about how green roof fraction is low compared to policy proposals and available space. However, it is important to remember, and maybe something to expand upon, that green roofs are not the only roofing strategy for heat mitigation. Of note, white roofs are generally found to be more effective that green roofs for reducing temperature, and solar panels, which have also become quite popular, can generate energy for indoor cooling. These alternative strategies would be competing for that same space.

**Thank you, this is a good point. We now mention this in the introduction.**

*On the other hand, green roofs impose an structural loads and additional costs, so are not always appropriate [cite FLL]; in other cases solar panels or high-albedo roofs may be more appropriate.*

---

## Author Response (AR2)

**Manuscript essd-2022-259, Second Revision**

February 3, 2023

We thank the editors and reviewers for their helpful comments and appreciation. They have lead to substantial improvements in our study. Please find point-by-point responses below. *Our responses look like this*, quotes from the changed text look like this.

Since the last revision, newer aerial imagery covering London became available to us. We therefore updated our work to use the newer imagery, and use the older imagery as a point of comparison for unseen imagery. Furthermore, in response to the reviewers we have implemented four-fold cross-validation. Together these are substantial changes to the dataset, and the results tables have changed completely; we include the new results tables in this letter.

By using four-fold cross-validation, as well as testing against imagery from a different year, we have fully addressed the referees' questions about generalisation of the model to unseen imagery.

**1 Referee 1**

I thank the authors for providing detailed responses to my original comments. I still have concerns about the inability for the trained CNN to generalise to new images. I believe the major corrections I recommended last time, relating to demonstrating the ability for the tool to generalise to detect green roofs in previously unseen imagery, is still necessary to accept this publication.

I agree with the authors that it is not expected that this tool would be able to detect green roofs in different types of imagery e.g., satellite imagery with a coarser spatial resolution. However, to demonstrate that this paper and tool provides a significant advance, further evidence is required to demonstrate that the tool can at least generalise to detect green roof area in aerial imagery captured by the same or similar sensors at similar times of the year. If this is not possible, it is not clear what the purpose and benefit of training a CNN over other methods and datasets, such as those cited in this manuscript (e.g., LRW2019 and GLA).

*Since the previous revision, new aerial imagery of London became available to us, collected in 2021. We therefore updated our study to use the 2021 imagery as its primary dataset, and used imagery from 2019 as an alternative test dataset. When the model trained on the 2021 dataset was tested against the 2019 dataset, precision was lower and recall higher than when tested against the 2021 dataset. However, we also tested performance of a model trained on the 2019 dataset, and found that it performed worse than the model trained on the 2021 dataset even when tested against the 2019 dataset. This demonstrates that the model was able to generalise to unseen imagery.*

*As a result of the change of dataset, all of the performance statistics have changed. Furthermore, in response to Referee 2 we applied four-fold cross-validation. The results shown in the tables are now the average across folds, whereas before we used a single split. Test data is still completely unseen during training. We therefore include the updated results tables in this letter.*

*By showing that performance is consistent between training splits through cross-validation, and by testing on completely unseen imagery from a different year, we demonstrate the ability of the*

*model to generalise.*

*In Section 2.1 we describe the two imagery sets.*

[revised manuscript text omitted]

It is recognised as a strength to this manuscript that the dataset is being made publicly available and that it covers the entire Greater London area, but if these are the primary novel factors in the paper, then it is at odds with the text which focusses primarily on the development and training of the CNN. Further emphasis has to be placed in the discussion section on what the repercussions of deriving such a dataset is. What is the benefit of having this information over and above what was provided in the previous datasets? Section 4.3. discusses the differences between the figures contained within the different datasets, but little information is provided on the repercussions of this.

*In order to emphasise the additional value provided by this dataset, we have added some detail to the discussion section covering the implications of the differing estimates. We have also added a new Section 4.5 which discusses the additional utility of this dataset through providing data at the level of single buildings. The previous studies do not make available their detailed geospatial results, only district-level averages.*

*In Section 4.3 we compare our district-level estimates with the results of earlier studies.*

Our estimate of green roof area in the CAZ in 2021 ($2.3 \times 10^5$ $m^2$) is higher than the LRW2019 estimates and the AMR estimate for 2013 and 2015, but lower than the AMR estimates for 2017. For Greater London, the identified area is higher than the 2016 and 2017 estimated areas from LRW2019. While individual-building data from previous studies are not available for comparison, local-authority district (LAD) level data are available from for 2017 from LRW2019 [Livingroofs Enterprises Ltd(2019), European Federation of Green Roof and Green Wall Associations (EFB) and ] In Figure 10, we compare our results for 2019 with the estimates for each LAD in 2017 from LRW2019: the results are strongly correlated, but some LADs have quite different results. According to this, most LADs have gained some green roof between 2017, with a few losing some. Newham (Nwm) and Hillingdon (Hdn) appear to have gained the most green roofs between 2017 and 2019. Our estimate for Havering (Hvg) is close to zero, because the 2019 imagery does not cover Havering (see Figure 2). Where estimates are differ by a small amount it may be due to differences in methodology or errors rather than a real change.

*In Section 4.5 we discuss the use of the new dataset, and give an example of the kind of analysis that can be performed.*

The dataset provides far greater detail than is available publicly from previous work in London. Green roof polygons are provided for individual areas of green roof, and are identifiable for individual buildings. This will enable new insights into the distribution of green roofs in London which were not possible before. For example, using the building use classifications given by the UKBuildings dataset, we can calculate the distribution of green roofs between building uses. As shown by Figure 11, non-residential buildings make up most of the buildings with green roofs (56%), with around 1.2% of non-residential building footprint area covered by green roofs compared to 0.3%

[Figure]

Figure 10: Scatter plot showing estimated green roof area in LADs of Greater London, comparing the estimates from [Livingroofs Enterprises Ltd(2019)] (2017) to our estimates (2019).

[Figure]

Figure 11: Share and proportion of green roofs for different building uses. Most green roofs are on non-residential buildings. Mixed use refers to buildings comprising both residential and non-residential uses.

of residential buildings. While a large fraction of green roofs occur on residential buildings, only a small proportion of residential buildings have a green roof. This illustrates the utility that this level of detail brings. Future work will extend this analysis to look in detail at the characteristics of buildings that have green roofs in London.

**2    Referee 2**

Summary: In the revised manuscript, the authors have clarified many things. My concern, however, remains the potential issues arising from the use a single training/validation split.

Major comment:

1. The major concern I have is the lack of cross-validation across training sets. The current model is constrained to work for the chosen training data and it is unclear how the feature selection would be impacted by a different training set. Some kind of cross-validation (Monte Carlo or k-folds) should at least be performed post hoc to confirm the ability of the model to generalize across training splits.

*In response to this comment we have applied four-fold cross-validation. Models were trained on four different train-test splits. Generalisation was also tested against unseen imagery collected in a different year. Please see the answer to Referee 1 for details.*

Minor comment:

1. colour (red, green, blue) raster images sounds strange. Maybe say rasters with red, green, and blue bands.

2. ' green roofs impose an structural loads and additional costs,' no 'an'

*We have corrected these mistakes.*

---

## Author Response (AR3)

Dear editor,

We are very pleased to see that our manuscript is accepted. The only correction requested was that we provide latitude and longitude markers on the region-of-interest map. We have done so, see Figure 1. We also noticed that the production file upload requires all multi-panel figures to be merged, so we have done so. There are no other changes to the manuscript.

Yours faithfully,

Charles Simpson, on behalf of the authors